# Vision-Based Autonomous Vehicle Systems Based on Deep Learning: A Systematic Literature Review

**Monirul Islam Pavel** , **Siok Yee Tan \*** and **Azizi Abdullah**

Center for Artificial Intelligence Technology, Faculty of Information Science and Technology,
Universiti Kebangsaan Malaysia, Bangi 43600, Malaysia; pavel@ieee.org (M.I.P.); azizia@ukm.edu.my (A.A.)
\* Correspondence: esther@ukm.edu.my

**Abstract:** In the past decade, autonomous vehicle systems (AVS) have advanced at an exponential rate, particularly due to improvements in artificial intelligence, which have had a significant impact on social as well as road safety and the future of transportation systems. However, the AVS is still far away from mass production because of the high cost of sensor fusion and a lack of combination of top-tier solutions to tackle uncertainty on roads. To reduce sensor dependency and to increase manufacturing along with enhancing research, deep learning-based approaches could be the best alternative for developing practical AVS. With this vision, in this systematic review paper, we broadly discussed the literature of deep learning for AVS from the past decade for real-life implementation in core fields. The systematic review on AVS implementing deep learning is categorized into several modules that cover activities including perception analysis (vehicle detection, traffic signs and light identification, pedestrian detection, lane and curve detection, road object localization, traffic scene analysis), decision making, end-to-end controlling and prediction, path and motion planning and augmented reality-based HUD, analyzing research works from 2011 to 2021 that focus on RGB camera vision. The literature is also analyzed for final representative outcomes as visualization in augmented reality-based head-up display (AR-HUD) with categories such as early warning, road markings for improved navigation and enhanced safety with overlapping on vehicles and pedestrians in extreme visual conditions to reduce collisions. The contribution of the literature review includes detailed analysis of current state-of-the-art deep learning methods that only rely on RGB camera vision rather than complex sensor fusion. It is expected to offer a pathway for the rapid development of cost-efficient and more secure practical autonomous vehicle systems.

**Keywords:** autonomous controlling; deep learning; decision making; intelligent vehicle; perception; self-driving

## 1. Introduction

Recently, the autonomous vehicle system (AVS) has become one of the most trending research domains that focus on driverless intelligent transport for better safety and reliability on roads [1]. One of the main motives for enhancing AVS developments is its ability to overcome human driving mistakes, including distraction, discomfort and lack of experience, that cause nearly 94% of accidents, according to a statistical survey by the National Highway Traffic Safety Administration (NHTSA) [2]. In addition, almost 50 million people are severely injured by road collisions, and over 1.25 million people worldwide are killed annually in highway accidents. The possible reasons for these injuries may derive from less emphasis on educating drivers with behavior guidance and poorly developed drivers' training procedures, fatigue while driving, visual complexities, that is, human error, which can be potentially solved by adopting highly efficient self-driving vehicles [3,4]. The NHTSA and the U.S. Department of Transportation formed the SAE International levels of driving automation, identifying autonomous vehicles (AV) from 'level 0' to the 'level 5' [5], where levels 3 to 5 are considered to be fully AV. However, as of

2019, the manufacturing of level 1 to 3 vehicle systems has been achieved but level 4 vehicle systems are in the testing phase [6]. Moreover, it is highly anticipated that autonomous vehicles will be employed to support people in need of mobility as well as reduce the costs and times of transport systems and provide assistance to people who cannot drive [7,8]. In the past couple of years, not only the autonomous driving academic institutions but also giant tech companies like Google, Baidu, Uber and Nvidia have shown great interest [9–11] and vehicle manufacturing companies such as Toyota, BMW and Tesla are already working on launching AVSes within the first half of this decade [12]. Although different sensors such as radar, lidar, geodimetric, computer views, Kinect and GPS are used by conventional AVS to perceive the environment [13–17], it is indeed expensive to equip vehicles with these sensors and the high costs of these sensors are often limited to on-road vehicles [18]. Table 1 shows a comparison of three major vision sensors based on a total of nine factors. While the concept of driverless vehicles has existed for decades, the exorbitant costs have inhibited development for large-scale deployment [19]. To resolve this issue and build a system that is cost efficient with high accuracy, deep learning applied vision-based systems are becoming more popular where RGB vision is used as the only camera sensor. The recent developments in this field of deep learning have accelerated the potential of profound learning applications for the solution of complex real-world challenges [20].

**Table 1.** Comparison of vision sensors.

| VS | VR | FoV | Cost | PT | DA | AAD | FE | LLP | AWP |
|---|---|---|---|---|---|---|---|---|---|
| **Camera** | High | High | Low | Medium | Medium | High | High | Medium | Medium |
| **Lidar** | High | Medium | High | Medium | High | Medium | Medium | High | Medium |
| **Radar** | Medium | Low | Medium | High | High | Low | Low | High | Low |

VS = Vision Sensor, VR = Visibility Range, FoV = Field of View, PT = Processing Time, DA = Distance Accuracy, AAD = AI Algorithm Deployment, FE = Feature Engineering, LLP = Low-Light Performance, AWP = All-Weather Performance.

In this systematic review paper, a broad discussion and survey of the implementation of deep learning are applied to aspects of AVS such as vehicle detection (VD), traffic signs and light identification (TSL), pedestrian detection (PD), lane detection and tracking (LDT), traffic scene analysis (TSA), decision making (DM), end-to-end controlling and prediction (E2EP), path and motion planning (PMP) and augmented reality-based HUD (ARH) analyzing research articles from 2011 to 2021 research articles on deep learning-applied AVS to reduce the dependency on sensor fusion and the high cost of manufacturing and to enhance the focus on developing a level 5 autonomous driving vehicle. We represent and thoroughly discuss the best deep learning algorithms for each domain, provide solutions to their limitations and analyze their performance for increasing practical implementation concepts. Moreover, this systematic review explored the most complete and predominate domains compared to other surveys from [21–33] (shown in Table 2) that indicates its impact on AVS implementing deep learning where the review article covered all aspects of the human–machine interface (HMI). The overall contribution of the research is set out below:

- Analyzed recent solution of state-of-the-art deep learning algorithms for cost-efficient AVS using RGB camera.
- Detailed literature review covering major domains and most subcategories to decrease vision sensor complexities.
- Discussed the key advantages and disadvantages of deep learning methods applied to AVS.

**Table 2.** Comparison of existing studies.

| Ref. | Year | Survey Coverage | | | | | | | | | |
|------|------|----|------|----|-----|------|-----|-----|----|-----|-----|
|      |      | VD | LRCD | PD | TSL | E2EC | TSA | PMP | DM | ARH | HMI |
| [21] | 2019 | ✔ | X | ✔ | X | X | X | ✔ | ✔ | X | ✔ |
| [22] | 2020 | X | X | X | ✔ | X | ✔ | ✔ | ✔ | X | X |
| [23] | 2016 | X | X | X | X | ✔ | X | ✔ | X | X | X |
| [24] | 2020 | X | X | X | X | ✔ | X | ✔ | X | X | ✔ |
| [25] | 2018 | X | X | X | X | ✔ | X | ✔ | X | X | X |
| [26] | 2018 | X | X | X | X | ✔ | X | ✔ | X | X | X |
| [27] | 2021 | X | X | X | ✔ | ✔ | X | ✔ | ✔ | X | ✔ |
| [28] | 2020 | ✔ | ✔ | ✔ | ✔ | ✔ | ✔ | X | X | X | ✔ |
| [29] | 2018 | X | X | ✔ | X | X | X | X | X | X | X |
| [30] | 2020 | ✔ | ✔ | ✔ | X | X | X | ✔ | X | X | X |
| [31] | 2020 | X | ✔ | ✔ | ✔ | ✔ | ✔ | ✔ | X | X | X |
| [32] | 2021 | X | ✔ | X | X | X | X | ✔ | ✔ | X | X |
| [33] | 2020 | ✔ | X | X | X | ✔ | ✔ | ✔ | X | X | ✔ |
| Ours | 2022 | ✔ | ✔ | ✔ | ✔ | ✔ | ✔ | ✔ | ✔ | ✔ | X |

## 2. Methodology

### 2.1. Review Planning

The study is based on a systematic review methodology, an approach for analyzing and evaluating accessible studies related to a particular issue of current research where the core three phases are preparing the review, conducting the review, and creating a report that summarizes the review. In this systematic review paper, the researchers have included 142 papers containing deep learning and belonging to a different domain of AVS. To finalize the papers, we initially focused on the entire domain of autonomous driving, then we restricted our search to the usage of deep learning in AVS. Only papers with full text in English from renowned journals, conferences and book chapters that were published between 2011 and 2021 were selected. Due to an increase in the scope of advanced autonomous transportation, we finally limited our search to the vision-based application of deep learning in AVS, and the rest were rejected. We also took the most complete edition to avoid dealing with duplication. The key plan and protocol of the review includes source of data, searching criteria and procedure, research questions, data selection and data extraction.

### 2.2. Sources of Data

Research papers were gathered from various famous research databases to incorporate specific field and research questions. Irrelevant research papers that could not address or endorse our research questions were dismissed. To achieve a broad coverage for the literature review, we used the following databases as our key resources: Web of Science, Scopus, IEEE Xplorer, ScienceDirect, MDPI, Springer, Wiley Library and ACM.

### 2.3. Research Questions

Research questions were formed to refine the survey and maintain the aim of the topic. The following research questions are answered throughout the discussion in the different sections of the paper.

- How does deep learning reduce sensor dependency?
- How are on-road objects detected and localized?
- What decision-making processes are solved for AVS?
- How does deep learning contribute to end-to-end controlling and path planning?
- How should final outcomes be represented in AR-HUD?

### 2.4. Searching Criteria

To find research papers according to the methodology, a pattern was followed to gather suitable papers which were mostly necessary for our study. We adopted a Boolean searching method with multiple AND, OR in the advance search options of each data source. During the search for the relevant papers, we selected "Autonomous Driving" and "Autonomous Vehicle" or "Intelligent Vehicle" or "Self-Driving" and "Deep Learning" as the main phrases. For a further refined search, various keywords were included to obtain the desired research papers according to our aim in this review. The following queries were developed based on Boolean operations:

- ((Autonomous Driving) OR (Autonomous Vehicle) OR (Intelligent Vehicle) OR (Self-Driving) AND (Deep Learning) AND (Object) AND ([Vehicle] OR [Pedestrians] [Traffic Sign] AND [Traffic Light]))
- ((Autonomous Driving) OR (Autonomous Vehicle) OR (Intelligent Vehicle) OR (Self-Driving) AND (Deep Learning) AND ([Traffic Scene] OR [Localization] OR [Segmentation]))
- ((Autonomous Driving) OR (Autonomous Vehicle) OR (Intelligent Vehicle) OR (Self-Driving) AND (Deep Learning) AND (Lane) AND ([Track] OR [Shift] OR [Segmentation]))
- ((Autonomous Driving) OR (Autonomous Vehicle) OR (Intelligent Vehicle) OR (Self-Driving) AND (Deep Learning) AND (Control) AND ([Steering] OR [Motion]))
- ((Autonomous Driving) OR (Autonomous Vehicle) OR (Intelligent Vehicle) OR (Self-Driving) AND ([Deep Learning] OR [Deep Reinforcement Learning]) AND (Decision Making) AND ([Uncertainty] OR [Lane Keeping] OR [Overtaking] OR [Braking] OR [Acceleration]))
- ((Autonomous Driving) OR (Autonomous Vehicle) OR (Intelligent Vehicle) OR (Self-Driving) AND (Deep Learning) AND ([Augmented Reality] AND [Head Up Display] OR [HUD]))

### 2.5. Searching and Extraction Procedure

The selection procedure for choosing papers includes four core iteration filtering processes. As the aim of the study is to discuss implementation of deep learning and comprehensive literature searches to analyze the frameworks and system designs, first, a total of 760 papers were selected from eight data sources based on the queries mentioned in the searching criteria (Section 2.4). Web of Science had the highest 151 and ACM had the lowest 40 papers. Then, the selected papers had to be processed through an eligibility stage where 209 duplicated papers were eliminated at first.

Furthermore, 121 papers were screened out during abstract scanning and 276 papers were chosen after full text reading. In the next iteration, studies containing domains of deep learning in relation to AVS were selected where all the papers were published between 2011 and 2021. The final dataset contains a total of 142 papers that covers the literature on the implementation of deep learning methods for AVS. The structure of the whole selection process is presented in Figure 1. Table 3 presents the final calculation for the selection of 142 papers according to these steps and based on the most relatable topics and in-depth analysis.

### 2.6. Analysis of Publication by Year

Out of 142 final papers for review, the studies published between 2011 and 2021 were selected. The year 2019 had the highest number of selected research papers, with 31, which is 17.9% of the total, and 2011 had the lowest number of papers (2). The distribution of publication is visualized in Figure 2.

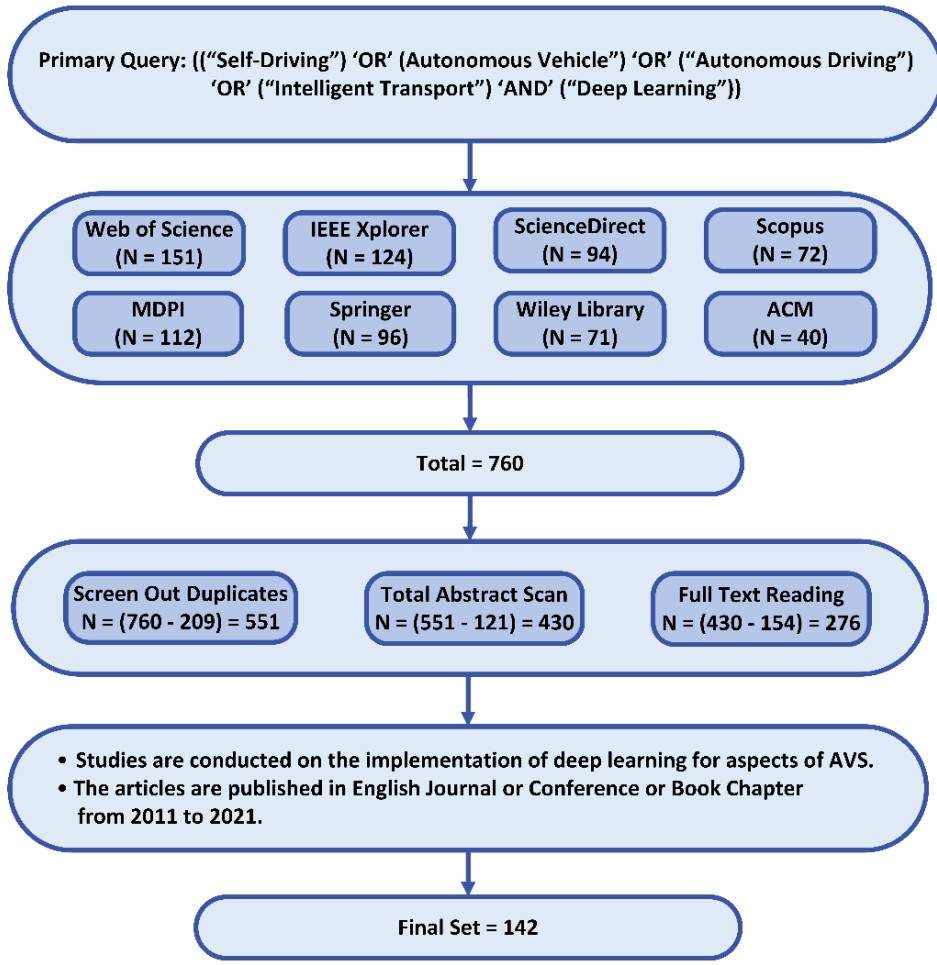

**Figure 1.** Framework for searching the literature and making the selection.

**Table 3.** Paper selection from multiple sources.

| Scheme 151 | Primary | Candidate | Selected |
|---|---|---|---|
| Web of Science | 151 | 72 | 24 |
| IEEE Xplore | 124 | 69 | 22 |
| Scopus | 72 | 47 | 20 |
| ScienceDirect | 94 | 51 | 19 |
| ACM | 40 | 32 | 7 |
| Springer | 96 | 53 | 21 |
| Wily Library | 71 | 44 | 8 |
| MDPI | 112 | 62 | 21 |
| Total | 760 | 430 | 142 |

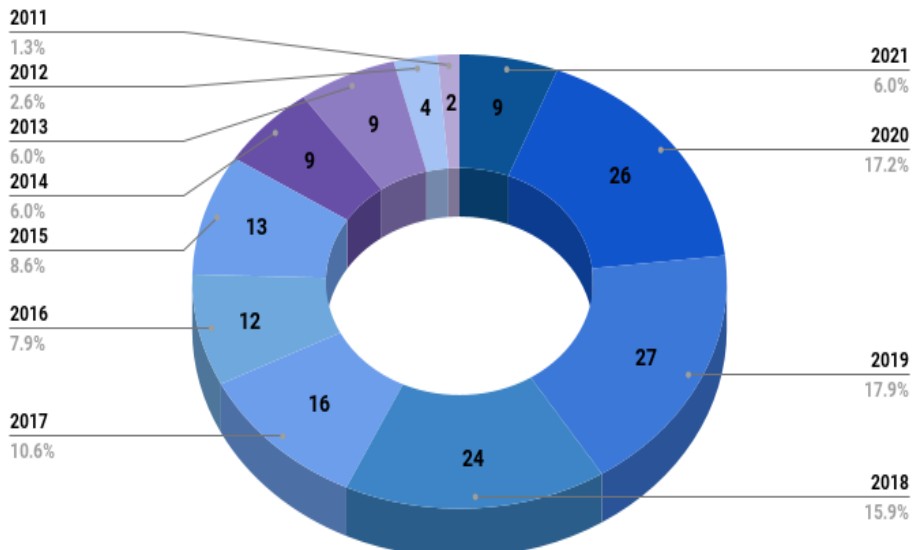

**Figure 2.** Distribution of studies in terms of year of publication (2011–2021).

*2.7. Analysis of Publication by Country*

Among the 142 selected papers for the literature review, 56 countries contributed to autonomous vehicle system development. Figure 3 shows the top 10 countries and the number of papers they contributed before the final selection. The graphical representation shows that China made the largest contribution, with 34 papers, and the USA contributed 21 papers, which was the second largest.

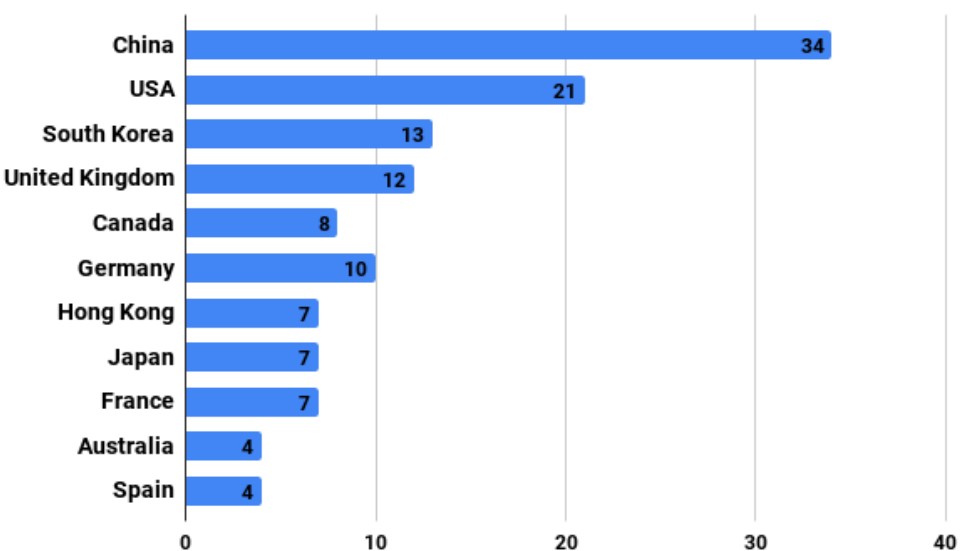

**Figure 3.** Distribution of studies over top 15 countries of first authors.

*2.8. Analysis of Publication Domains*

The 142 final papers were selected based on five domains and five subdomains of perceptions, shown in the literature taxonomy of AVS in Table 4, which were combined to produce a complete system. Table 4 shows that the distribution of 'Decision Making' section has highest 20 papers, and 'Path and Motion Planning' and 'AR-HUD' have the lowest 11 papers individually.

**Table 4.** Literature taxonomy of AVS using deep learning approaches.

| Domain | Sub-Domain | References |
|---|---|---|
| Perception | Vehicle Detection | [34–45] |
| | Traffic Sign and Light Recognition | [46–59] |
| | Pedestrian Detection | [60–78] |
| | Lane Detection and Tracking | [44,79–101] |
| | Traffic Scene Analysis | [55,102–120] |
| Decision Making | - | [121–143] |
| End-to-End Controlling and Prediction | - | [144–163] |
| Path and Motion Planning | - | [164–175] |
| AR-HUD | - | [176–186] |

To visualize the leading algorithms of each domain or subdomain, Figure 4 presents the distribution of algorithms, where the reviewed algorithm-centered approaches have a predominant role in AVS development. Figure 5 shows the dataset clustering which was used for the reviewed approaches. Only the subdomains of perception showed dependency on dataset, where "Traffic Sign and Light Recognition" and "Lane Detection and Tracking" applied to 6 datasets each, and only 3 datasets were adopted in "Traffic Scene Analysis".

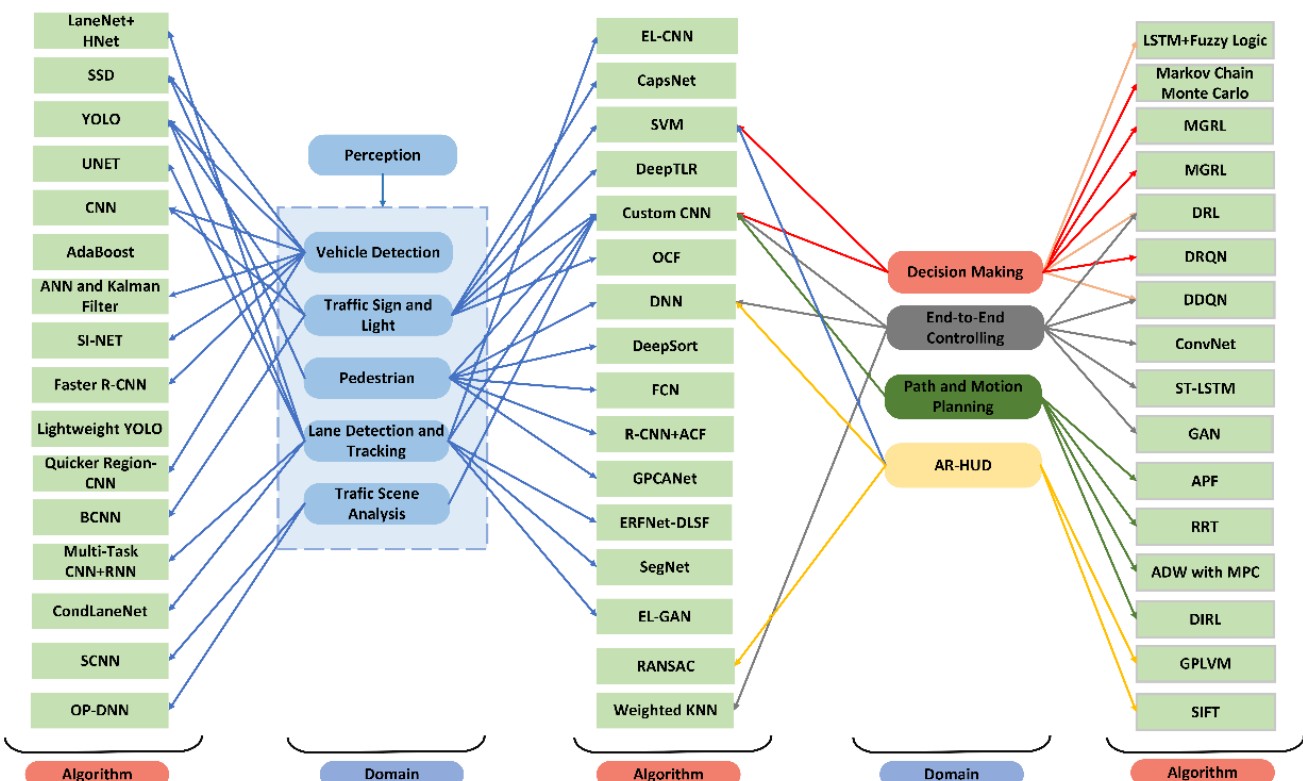

**Figure 4.** Taxonomy algorithms for each domain.

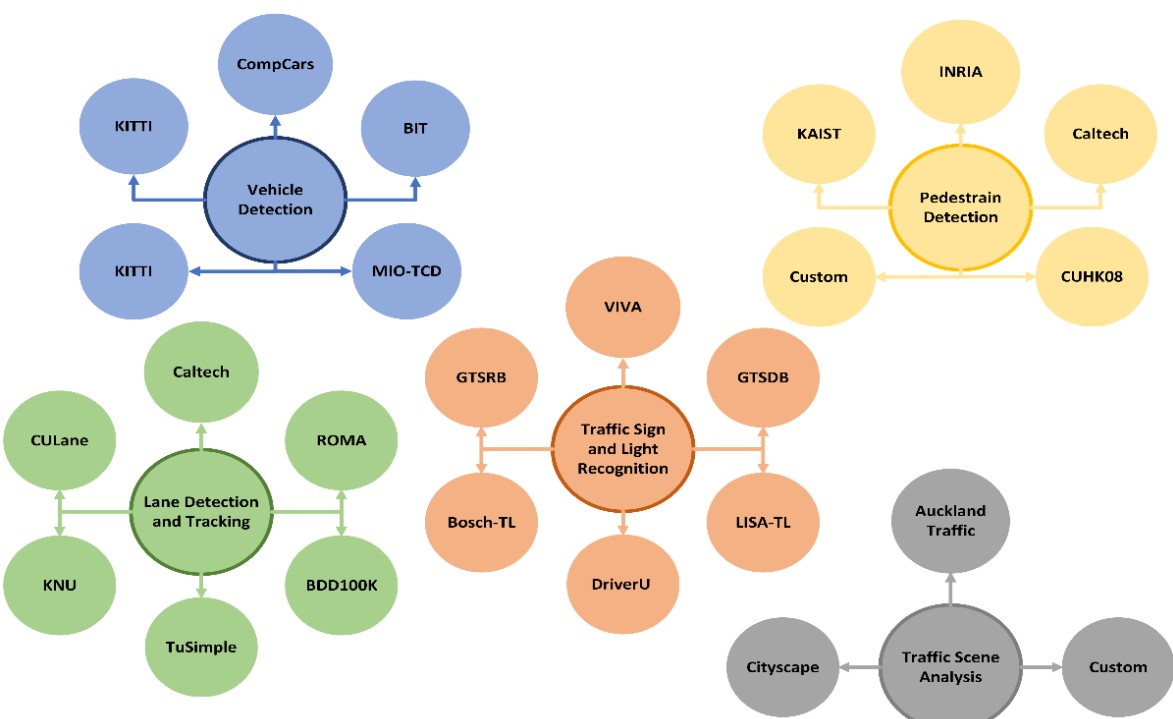

**Figure 5.** Clustering of dataset for subdomains of perception.

## 3. Analysis of Domain

Each domain was analyzed by reviewing several approaches and methods based on evaluating and discussing advantages, disadvantages, outcomes and significance. The following analysis of each domain was carried out with the aim of accelerating development of level 4 or 5 AVS.

### *3.1. Perception*

#### 3.1.1. Vehicle Detection

The identification and detection of an on-road vehicle for AVS together form one of the predominant and most challenging issues due to versions, combined fast multitasking and visual difficulties. For fast and more accurate vehicle detection and recognition in different and uncertain driving conditions, deep learning algorithms are analyzed in this section.

For instance, an online network framework for detecting and tracking vehicles was proposed by Hu et al., who predicted full 3D vehicle bounding box mapping from a monocular camera using both the environment and camera coordinates by reprojecting [34]. Eventually, the framework tracked the movement of instances in a global coordinate system and revised 3D poses with a trajectory approximation of LSTM, implementing on a KITTI dataset, where the outcomes surpassed the outcomes of LiDAR in long range [187]. In a 30 m range, LiDAR obtained 350.50 false negative and the method scored 395.33, while vehicle detection was 11.3% higher, indicating the limitation of the framework. However, it performed better in 50 m and 10 m, where the false negative scores were 857.08 and 1572.33 when the LiDAR-based method obtained false negative values of 1308.25 and 2445.30, respectively. The decreased false negative in 10 and 50 m showed that the method was able to overcome the performance of LiDAR using only camera and deep learning despite reduced accuracy in some real-time implementations.

To tackle accuracy-based issues, improve slow detection and recognition speed, and address the lack of categorization ability, Sang et al. introduced a novel YOLOv2_Vehicle architecture [35]. For multiple scales of vehicles that influenced the detection framework, normalization had been used for the improvement of the method of measuring losses for boundary box length and width after clustering and bounding boxes applying k-mean

++ algorithm to the training dataset [36] along with applying multilayer feature fusion to boost the network extraction capabilities and repeatedly eliminating convolutional layers in high layers. The method implementing the BIT dataset could obtain a mean average precision (mAP) exceeding 94.78% in 0.038 s, which was found to be much faster and more accurate than compared existing methods.

In another work, an AdaBoost combined with a pixel look-up features-based approach was demonstrated by Ohn-Bar et al., where the methods included mining orientation, object geometry and occlusion pattern by clustering, and 81.94%, 66.32%, 51.10% accuracy in easy, moderate and hard scenarios, respectively, was obtained for vehicle detection. However, performance decreased when a 70% overlap evaluation threshold was used instead of 50% and during heavy occlusion [37]. Nonetheless, the method was inappropriate for rough conditions as performance decreased when 70% overlap evaluation threshold was used instead of 50%, and it showed poor accuracy in heavy occlusion.

Further, Chen et al. presented a new method to identify five distinct vehicle classes, that is, car, van, bus, truck and tractor, using the AdaBoost and CNN algorithms applied to CompCars and their custom dataset containing rear views of vehicles [38]. They employed CNN as a function extractor with a Support Vector Machine (SVM) for training the features separately, and further AdaBoost algorithms were applied for integration. They obtained optimum results even with faulty images and high computing costs, with average accuracy of 99.50% in 0.028 s, which was 13% higher than the other mentioned fusion methods, for instance, SIFT + SVM, HoG + SVM and SURF + SVM. However, the method was only deployed and considered in simple but low-quality images and daytime scenarios.

Moreover, one of the biggest issues was the low resolution of images in the real-time traffic surveillance method due to either low-vision RGB cameras or environmental features such as low light condition or foggy weather. For this problem, vehicles in low-resolution images and videos were analyzed in terms of the efficiency of the CNN by Bautista et al. [39]. The neural network used an activation function that worked in two phases: first, the detection of high-level attributes; second, the detection of low-level attributes. It tested the comportment of the model to detect vehicles with lower input resolution at different levels, as well as the number and size of filters. Results demonstrate that CNN was remarkably successful even with low resolution in the identification and classification of vehicles with an average precision fit for real time applications.

Lee et al. showed a hierarchical system for detecting and tracking vehicles in an urban area at night based on taillights [40]. The system focused primarily on effective detection and pairing of taillights, considering their innate variety and observing all aspects of the layers and interrelationships in a hierarchical framework which increases the efficiency of vehicle detection and tracking in comparison with traditional methods, with recall of 78.48% and precision of 90.78%. However, performance decreased for short-distance vehicles due to headlight illumination of host vehicles. This approach could be considered as one the most suitable methods for nighttime vehicle detection.

Hu et al. demonstrated a novel CNN architecture called scale-insensitive convolutional neural networking (SI-Net) [41] to enhance the performance of vehicle detection for autonomous vehicles, solving the issue of limited CNN-based vehicle detection [39]. The framework improved the limitation-scale insensitive CNN, deploying context-aware region of interest (ROI) pooling to preserve the real structure of small-scale objects. The state-of-the-art method outperformed the others in terms of measuring, scoring 89.6%, 90.60% and 77.75% for accuracy in moderate, easy and complex moods, respectively, in 0.11 s execution time on the KITTI benchmark as well as a custom highway dataset with different variance of scaled objects. Thus, the method was able to maintain a good performance in multiple traffic scenarios.

Targeting the runtime of previous works, Wang et al. combined anchor size, receptive field and anchor generation optimization (AGO) with Fast R-CNN to ensure that an acceptable number of vehicle features could be accessed by the network in the shortest amount of time [42]. Using the anchor shape, it efficiently detects vehicles in large, medium

and short fields of view with 87.2% average precision in 0.055 s. The anchor shape-based detection process is a very coherent technique for AVS for reducing computational cost by not taking the whole field of vision for processing.

In another work, which combined Faster R-CNN training parameters with a region proposal network (RPN)-based approach, Suhao et al. implemented vehicle-type detection in a real traffic area including MIT and Caltech datasets with ZF and VGG-16 networks in multiple scenarios [43]. The research results increased the average accuracy of the detection systems and the rate of detection compared with the CNN models. The proposed architecture classified vehicles from three categories where they achieved the best accuracy: 84.4% for car, 83.8% for minibus and 78.3% for SUV using the VGG-16 model in 81 ms. The execution cost of the proposed method outperformed Fast R-CNN and Faster R-CNN applied to complex scenarios.

However, a lightweight YOLO network that was built with a YOLO v3 algorithm with generalized IoU loss was combined with loss function as well as with the integration of two different focal length cameras by Liu et al. to indicate less computer complexity for AVS [44]. The method was implemented on their self-made dataset where the network obtained 90.38% precision and 82.87% recall within 44.5 ms. This could be a milestone for AVS in terms of a faster and more accurate method for different field of view and day or nighttime implementation.

Leung et al. compared deep learning-based techniques for vehicle detection efficiency [45], and proposed solutions for data collection along with the nighttime data labelling convention to resolve different types of detection. The research also recommends a framework based on a quicker region-based CNN model, which was precisely optimized, merging with RestNet101, the VGG-16 model obtaining a mean average precision (mAP) of 84.97%. The experimental result showed a high detection accuracy in urban nighttime with version lighting conditions including extreme low light and no lighting. Thus, this method became one of the most suitable methods for AVS in challenging lighting conditions.

Overall, the Deep CNN and AdaBoost-based approach achieved 99.50% accuracy in daytime (the highest) with the fastest computational time (0.028 s), but lightweight YOLO and the quicker region-based CNN model showed practical outcomes in both daytime and nighttime scenarios for vehicle detection. Multiple deep learning methods showed efficient performance by improving slow detection, recognition and categorization, enabling deployment in complex scenarios and night-time deployment with good accuracy, even surpassing accuracy outcomes of LiDAR in terms of long field of view. However, some challenges remained, for example, limited dataset of vehicle categories, performance dropping in low light and rough weather conditions for some methods, low accuracy for vehicle detection in short distance for headlight illumination at nighttime and fast execution time in real-time implementation. An overview of the methods evaluated for the detection and recognition of vehicles for AVS is provided in Table 5.

### 3.1.2. Traffic Sign and Light Recognition

One of the most important aspects of a safe and better decision-making process for automotive driving system was traffic sign and light identification by regulating traffic, monitoring and avoiding accidents through warning the drivers. Traffic sign and light recognition systems follow a double-step process, detection and classification, where detection denotes correctly spotting the geometric position in the image and classification means identification of the category in which the detected sign or light signal appears [28,188].

A bio-mechanism inspired novel architecture named Branch Convolutional Neural Network (BNN) was proposed by Hu et al. for traffic sign recognition [46]. To improve the recognition machine speed and accuracy, a branch-output mechanism which was placed between pooling and convolutional layer and added to the framework. Furthermore, instead of the initial output layer, the sign in the preceding branch was projected by the BCNN that results perfect prediction in partial visibility of road signs with 98.52%

accuracy based on German Traffic Sign Recognition Benchmark (GTSRB). For complex visual scenarios, BCNN based approach worked very well for traffic sign recognition.

**Table 5.** Summary of multiple deep learning methods for vehicle detection.

| Ref. | Method | Outcomes | Advantages | Limitations |
|------|--------|----------|------------|-------------|
| [34] | 3D vehicle bounding box mapping | 65.52% more enhancement in 50 m than LiDAR. | Exceeded the outcomes of LiDAR in long range. | Unstable accuracy for certain error properties. |
| [35,36] | YOLO V2 and K-mean ++ | mAP 94.78% in 0.038 s. | Faster detection and recognition. | Trained with vehicle types of data. |
| [37] | AdaBoost with pixel lookup features | 81.94% accuracy in best-case scenarios. | Improved result with occlusion pattern by clustering. | Performance decreased when 70% overlap and heavy occlusion. |
| [38] | Deep CNN and AdaBoost | 99.50% accuracy in 0.028 s. | Highest accuracy in daytime and trained with low-quality frames. | Not applicable in low-light or complex scenarios. |
| [39] | CNN with Caffe framework | 96.47% accuracy in 51.28 ms. | Fast classification in low resolution input. | Higher execution time compared to other methods. |
| [40] | ANN and Kalman Filter | Recall 78.48% and precision 90.78% in urban scenario. | Nighttime taillight-based detection in urban area. | Decreased in performance for headlight illumination in short distance. |
| [41] | SI-NET | 90.60% accuracy in best-case scenarios. | Improved the limitation-scale insensitivity of CNN. | Not applicable in poor lighting, nighttime or complex weather scenarios. |
| [42] | Faster-RCNN with AGO | 87.2% average precision in 0.055 s. | Enable to detect vehicle in large, medium and short field. | Not applicable in urban or challenging lighting conditions. |
| [43] | Faster R-CNN with RPN | Highest 84.4% accuracy for car detection in 81 ms. | Outperformed Faster R-CNN in terms of execution time. | Unsatisfactory accuracy compared with the obtained execution time. |
| [44] | Lightweight YOLO Network | 86.81% and 78.91% precision and recall within 44.5 ms, respectively. | Applicable in both day and night scenarios in multiple FoV. | Was not tested in urban or crowded environment. |
| [45] | Quicker region-based CNN | 84.97% mAP nighttime scenarios. | Able to detect in low-light and almost no-light conditions. | Required huge manual data labelling. |

Jung et al. trained the 16 different forms of Korean traffic sign with LeNet-5 CNN architecture for a real-time traffic sign recognition system where the training set had 25,000 positive and 78,000 false samples [47]. The method obtained up to 99.95% within a fast-processing time. The applied color property-based CNN approach could be very efficient for lightweight traffic sign detectors for AVS as well as achieving the highest accuracy.

An improved traffic sign recognition algorithm was demonstrated by Cao et al. for an intelligent driving system [48]. For accurate detection spatial threshold segmentation, the HSV color space was utilized, and traffic signs were identified accurately depending on shape features and processed with LeNet-5 CNN architecture with Gabor kernel, which was the primary convolutional kernel, and batch normalization was applied after the pooling layer. The Adam model was also implemented as the optimizer algorithm. The proposed methodology was applied to the German Traffic Sign Recognition Benchmark and obtained 99.75% accuracy with 5.4 ms per frame on average, which was higher in both sectors than [189,190], where the accuracies were 98.54% in 22 ms and 95.90 in 5.4 ms, respectively, adopting HoG + PCA and multilayer perceptions methods.

On the other hand, the parallel architecture weighted multi-convolutional neural network took 4.6 ms more to process but still achieved constant high efficiency, scoring 99.75% in GTSDB and 99.59% accuracy in the GTSRB dataset, where low and complex lighted scenarios were also considered. Despite occasional accuracy drops for blurry vision, this method could be one of the most suitable approaches for AVS [49].

To detect traffic signs, Wang et al. suggested a red bitmap extraction and SVM-based method where the detected images were color-segmented and afterwards the shape detection of ROI (ROI) was carried out on the basis of rim detail [50]. The methodology scored recall values of 97% and 99% for danger and prohibitor for the GTSDB dataset, respectively. This technique obtained good detection accuracy, but the major limitation was that this method was only applied to red circular signs.

Zhang et al. demonstrated a modified YOLO V2 algorithm to develop an improved Chinese traffic sign detection system, as well as constructing the database [51]. In order to create a single convolutional network, they used several $1 \times 1$ convolutional layers for the intermediary and fewer convolutional layers for the top layers. Fine grid was also used to separate images with the goal of identifying small-sized road signs. Their technique was found to be the outcome of the CCTSDB and GTSDB databases, where AUC values for mandatory and danger signs were 96.81%, 94.02% and 96.12% in 0.017 s.

Another approach applied CapsNet, which resolved the major limitation of CNN, that is, loss of max pooling layer retaining spatial relations [52]. The approach obtained 98.72% accuracy for recognizing traffic light with shape. It can be a useful method-based approach for AVS's traffic sign-recognition methods.

Furthermore, a unified deep convolutional traffic light-identification feature for automated driving systems was proposed by Bach et al., based on Faster R-CNN that was suitable for detection of traffic lights, and the recognition and classification of types or states [53]. They achieved 92% average precision applying on a large-scale dataset named DriverU traffic light. When the width was greater than 8 px and smaller than these, it scored 93% for average precision. However, there were still limitations for suitable number of false positives which can be reduced by applying RNN or an integrated approach. DeepTLR was the real-time vision-dependent, in-depth and deeply convoluted traffic light-identification and classification system that did not require position details or temporal principles; these were proposed by Weber et al. [54].

On the basis of the single-frame assessment of a challenging collection of urban scenes, the authors presented noteworthy outcomes, showing that in regular images, DeepTLR achieves frame rates of up to 33 Hz. DeepTLR also ran at frame rates of up to 33 Hz. DeepTLR also ran at frame rates of 13 Hz on images with a resolution of $1280 \times 960$ pixels. The capacity for more transport lights was high in the architecture, scoring 93.5% F1 score for $1280 \times 960$ resolution and 88.1% F1 score for $640 \times 480$ in 80 ms and 28.8 ms.

Li et al. developed a framework of robust traffic-light recognition with fusion detection in complex scenes [55]. To increase accuracy for each traffic light type and the creation of a fusion detective framework, a set of enhanced methods was adopted based on an optimized channel function (OCF) system after using aspect ratio, field, location and traffic lights background as prior knowledge to minimize computational redundancy and create a task model for the identification of traffic light. Furthermore, they utilized the detection knowledge of the previous system to change the original ideas, which further increased the accuracy. The framework was applied to a VIVA dataset where a combination of multi-size detectors, bulb detectors and fuzzy detectors were implemented, which improved the AUC indicator, with 7.79% for red, 9.87% for red left, 11.57% for green and 3.364% for green left, compared with general ACF on VIVA validation dataset and achieved an AUC indicator of 91.97% for red light and 89.32% for green light on the channel-modified LARA validation dataset.

In addition, to reduce complexity, Lee et al. adopted the concept of upper-half clipping frame so that the model could pick only those frames that would allow it to recognize traffic lights rather than taillights [56]. The system was built based on a YOLO algorithm

and obtained 42.03% mAP and 49.1% mAP enhanced, and improved results applied to the Bosch-TL and LISA-TL datasets, but the author did not consider nighttime scenarios. Other than this issue, the method was exceptionally successful for traffic signs and light identification systems for AVS.

Behrendt et al. implemented YOLO for 3D localization and tracking of traffic lights [57]. A wide field of view was considered, and the YOLO-based approach was deployed in the Bosch-TL dataset. The method showed 99% accuracy in just 0.06 ms. However, the method required huge pre-labelled data which created an obstacle to fluent performance.

In this section, both traffic signs and traffic light detection and recognitions are discussed and summarized in Table 6, for which most of the deep learning approaches were trained with the GTSRB dataset. Among all the deep learning methods, LetNet-5-based CNN on self-made dataset with spatial threshold segmentation with the HSV color space and Gabor filter on the GTSRB dataset performed best for traffic-sign recognition despite reduction in performance in a complex environment and detection of separated signs due to the proposed region. In addition, the YOLO method-based approached obtained highest accuracy in the fastest time for traffic-light detection with recognizing inner signs despite having pre-labelled data dependency.

**Table 6.** Summary of multiple deep learning methods for traffic sign and light recognition.

| Ref. | Method | Outcomes | Advantages | Limitations |
|---|---|---|---|---|
| [46] | BCNN | 98.52% accuracy for low visibility of road signs. | Branch-output mechanism enhanced recognition speed and accuracy. | Implementation from moving platform was not tested. |
| [47] | LetNet-5 based CNN | 99.95% accuracy. | Lightweight color segmentation before classification, improved processing speed. | Detected two traffic signs for having different location of proposed region. |
| [48] | CNN with Gabor filter | Accuracy of 99.75% in 5.4 ms. | Obtained highest accuracy and enhanced the performance of LetNet-5. | Performance decreased in complicated backgrounds. |
| [49] | Weighted multi CNN | Highest 99.75% accuracy in 10 ms. | Classification in high-speed driving and outperformed in low-light conditions. | Struggled to classify in challenging blurry vision condition. |
| [50] | SVM | Recall score 97% and 99% for two cases. | Faster detection of prohibitory and danger signs in poor illumination. | Only applied to red circular signs. |
| [51] | YOLO V2 | Highest 96.81% AUC values in 0.017 s. | Faster and simple processing pipeline. | Decrease in performance for small traffic signs. |
| [52] | CapsNet | 98.72% accuracy. | Resolved loss of max pooling to boost CNN's performance. | Did not consider complex lighting condition |
| [53] | Fast R-CNN | 100% recall and 92% average precision. | Detected traffic light as well as indicating signs. | Showed high false positive. |
| [54] | DeepTLR | Highest 93.5% F1 score. | Did not require position details or temporal principles. | Lower precision rate. |
| [55] | OCF | 11.5% improved AUC value for green light. | Better detection in complex scenes with low luminous objects combining. | Unsatisfied accuracy for red left traffic light identification. |
| [56] | YOLO | 42.03% and 49.16% mAP higher on two datasets. | Took upper half of frame to eliminate search area and vehicle taillights. | Did not deploy in nighttime traffic scene where reflection creates confusion. |
| [57] | YOLO | 99% in 0.06 s. | 3D localization and tracking of traffic lights in large field of view. | Required more labelling. |

### 3.1.3. Pedestrian Detection

Effectively detecting and localizing pedestrians on roads in various scenarios was one of the major vision-based problems for autonomous driving systems. A study shows that only in the USA has the fatality rate for road crossing increased up to 30% in seven years from 2009. In 2016, a total of 6000 pedestrians were killed, which is a record in the last three decades [58]. Moreover, based on vulnerable road users in the ASEN region, 13% of the deaths on roads are related to pedestrians [59]. In order to prevent deaths, the detection and localization of pedestrians have become a major focus of the study of autonomous vehicles. Several studies have been successfully conducted on reducing accident cases and creating a sustainable and more accurate approach to autonomous driving systems.

For instance, Angelova et al. proposed a deep network named large-field-of-view (LFOV) to perform complex image processing continuously for pedestrian detection and localization [60]. The purpose of the proposed Large-Field-of-View deep network was to understand, simultaneously and effectively, as well as to make classification decisions in many places. The LFOV network processes vast regions at much higher speeds than traditional deep networks and therefore can re-use calculations implicitly. With 280 ms per image on the GPU and 35.85 on the average miss rate on the Caltech Pedestrian Detection Benchmark, the pedestrian detection system showed a promising performance for real-world deployment.

A vision based pedestrian detection and pedestrian behavior classification technique was proposed by Zhan et al. [61], where YOLOv3-TINY was used for quick segmentation and multitarget tracking of detected pedestrians with the DeepSort algorithm [62]. Finally, to identify the behavior of pedestrians, an improved and customized AlexNet algorithm was adopted. The proposed model performed efficiently in real time at a rate of 20 frames per second along with a designed warning area binding each pedestrian.

Convolutional neural network is one of the most popular deep learning models and has been adopted in several studies for pedestrian detection. Ghosh et al. used a novel CNN architecture model for pedestrian detection [63]. To train the model, they applied transfer learning as well as synthetic images using an uncovered region proposal of a bounding box to avoid the annotation of pedestrians' positions. It obtained a 26% missing rate in CUHK08 and a 14% missing rate in the Caltech pedestrian dataset, where crowded scenes were considered. The biggest advantage was that it required no explicit detection while training and did not need any region proposal algorithm.

A similar concept was used by Wang et al., who combined part-level fully convolutional networks (FCN) and CNN to generate a confidence map and pedestrian location based on the aligned bounded box concept [64]. The proposed framework was compared with CifarNet and achieved 6.83% improved outcomes.

A novel single shot detector method based on late-fusion CNN architecture was introduced by Hou to analyze data of a multispectral system that performed with higher accuracy at nighttime [65]. The combined architecture was applied to a KAIST multispectral pedestrian benchmark where the late-fusion CNN architectures worked efficiently. In terms of log average miss rate, it decreased by more than 10% and developed for suitable deployment during both day and nighttime. As a result, it became one of the best practical CNN-based pedestrian detectors of all the accepted AVS methods.

For identifying pedestrians in low resolution learning from low-level image features, a single image-based novel resolution-aware CNN-based framework was proposed by Yamada et al. [66]. The authors also developed a multiresolution image pyramid and obtained the original input image to identify pedestrian size. Moreover, it learnt feature extraction from a low-level image with resolution information and achieved 3.3% lower log-average miss rate than CNN which made the architecture more acceptable for AVS.

In another work, Zhang et al. implemented an optimized multiclass pedestrian identification system, using a Faster RCNN-based neural network [67]. The analysis indicated that the framework for pedestrian detection in blurred fields of view were able

to increase speed with average precision of 86.6%. This approach could be suitable for distorted images for pedestrian detection tracking.

Dong et al. proposed a region proposal framework for pedestrian detection implementing R-CNN combined with an ACF model, where the ACF model was applied to produce only pedestrian class-bounding region, which was a very useful application for autonomous vehicle systems [68]. Moreover, the proposed framework cost less execution time during training and testing. Though most of the studies showed pedestrian detection in the daytime or in clear weather, this task becomes more complex in low-light condition, haze or fog because these create vision difficulties and this kind of condition causes a higher number of accidents [69,70] and increases the possibility of traffic accidents by 13% [71].

Correspondingly, de-hazing algorithms were one of the solutions to fix vision problems which can be implemented in detection of pedestrians in haze conditions. For instance, a related approach for pedestrian detection in haze conditions was proposed by Ding et al., implementing a synthesized haze version of the INRIA dataset using dark channel prior-based linear SVM and HOG algorithm [72]. Nonetheless, the approach received poor recall value, scoring 67.88% in predicting constant depths of input images. Although the method is a good approach for pedestrian detection, the limitations could be solved taking into account the pre-trained pedestrians' depths in multiple haze environments. Moreover, Huang et al. provided a Laplacian distribution model that featured a combined HTE (haze thickness estimation) and IVR (image visibility restoration) for solving problems of pedestrians [73]. Implementation of this algorithm could enhance the performance for detecting pedestrians in haze conditions, but the most difficult haze conditions occur in dim light conditions, where they achieved 4.1 mAP and 3.98 mAP based on expert and ordinary view, respectively.

For alleviating the haze problem while detecting pedestrians, Li et al. [74] proposed three approaches named Simple-YOLO, VggPrioriBoxes-YOLO and MNPrioriBoxes for pedestrian detection based on YOLO [75]. Deep separable convolution and linear bottleneck capabilities were implemented to minimize parameters and enhanced processing speed, making the network far more usable. The average precision of their three methods was 78.0%, 80.5% and 80.5%, respectively, where precisions were 89.4%, 90.8% and 89.3%. The lowest 22.2 FPS were 22.2, 81.7 and 151.9 applied to the combined data of their HazePerson dataset and INRIA person dataset after dataset augmentation. Although this approach was one of the preferable approaches for AVS to detect and localize pedestrians in day and night haze conditions, the higher missing rate in complex scenarios was an issue which could be resolved by adopting key point detection methods. Xu et al. proposed a ground plane context aggregation network (GPCANet) for detecting pedestrians in ground plane areas of Caltech, SCUT and EuroCity datasets, where the best result was achieved for the SCUT dataset with 96.2% recall value, and obtained 25.29% and 25.10% log average miss rate for the rest of the dataset serially [76]. However, it might have slightly higher log average miss rate, but the outcomes were in crowded traffic complex scenarios, which made the approach more practicable for AVS.

Moreover, CNN-based work was demonstrated with only 5.5% miss rate to localize distracting pedestrians [77]. Similarly, CNN cascaded with AdaBoost was deployed for pedestrians in night images [78]. It obtained a maximum 9% log-average miss rate, although both methods are not evaluated in complex scenarios.

In summary, multiple deep learning methods were reviewed (shown in Table 7) where a CNN-based method was deployed for faster pedestrian detection and localization where the methods showed 94.5% success rate and provided an improved dataset built on the Caltech dataset. FCN achieved 6.83% improved outcomes compared with CifarNet, while in terms of estimating distance of pedestrians from the vehicle, it showed a higher missing rate. Moreover, GPCANet performed best on the SCUT dataset, scoring 96.2% recall in 320 ms and deployed in diverse scenarios in both day and night conditions. However, it scored a high missing rate and could not deal with complex scenes in terms of occluded road objects. However, when of the other methods showed stable efficient outcomes,

the main challenges remained for crowded traffic scenes and complicated visual and weather conditions.

**Table 7.** Summary of multiple deep learning methods for pedestrian detection.

| Ref. | Method | Outcomes | Advantages | Limitations |
|---|---|---|---|---|
| [60] | DNN | 35.85 on the average miss rate. | Processed in large field images, continuous detection in complex scenes. | Higher missing rate with comparatively slow detection |
| [61] | YOLOv3-TINY and DeepSort | 80.3% accuracy in complex environment. | Designed faster warning area bounding by each pedestrian with direction labelling. | Only considered daytime scenarios and lower accuracy. |
| [63] | CNN | 26% and 14% missing rates on two datasets, respectively. | Did not require explicit detection in crowded scenario. | Did not apply for motion images or real-time problems. |
| [64] | Part-level FCN and CNN | 6.83% improved outcomes compared with CifarNet. | Estimated accurate distances of pedestrians generating the confidence map using FCN. | High missing rate for practical implementation. |
| [65] | SSD-based late-fusion CNN | Decreased by more than 10% of log average miss rate. | Most applicable in nighttime implementation. | Slower detection and complex parameter tuning. |
| [66] | Resolution aware-based CNN | 3.3% lower log-average miss rate than CNN. | Learnt feature extraction from low-level image with resolution information. | Was not applied in complex traffic or crowded environment. |
| [67] | Faster R-CNN | 86.6% average precision in 0.104 s. | Outperformed on distorted and blurry frames. | Did not consider low-light or traffic scenarios. |
| [68] | R-CNN with ACF model | 14.1%, 15.3%, 45.6% miss-rate on three datasets. | Reduced the number of region proposals, and costs less time. | — |
| [72] | Dark channel-based SVM | 81.63% precision and 67.88% recall. | Pedestrian detection and position estimation from haze condition. | Presumed constant depths in input images. |
| [73] | Laplacian distribution model | 4.1 mAP and 3.98 mAP on expert and ordinary view, respectively. | Pedestrian detection in complex dim-light condition. | Was not applied in real-time driving scenes. |
| [75] | Multiple YOLO methods | Highest average precision 80.5% in 81.7 ms. | Minimized number of parameters and outperformed state-of-art methods. | Early dependency on preliminary boxes during detection process. |
| [76] | GPCANet | 96.2% recall in 320 ms on SCUT dataset. | Improved outcomes in both day and night including far FoV and crowded traffic. | Higher log average missing rate for occluded on-road objects. |
| [77] | CNN | Showed 5.5% miss rate. | Localized distracting pedestrian and improved detection annotations. | Did not test in cases for crowded or complex scenes. |
| [78] | CNN cascaded with AdaBoost | Generated the maximum 9% log-average miss rate. | Combined thermal images for nighttime detection. | Might fail in complex urban traffic scenarios. |

### 3.1.4. Lane Detection and Tracking

One of the core fundamentals for AVS was to identify lane and tracking curves in real time where the controlling would depend on the lane and curves. Several studies have been conducted on this field based on different camera visions implementing deep learning and computer vision approaches considering color, texture, feature extraction in different scenarios for lane detection, lane shifting, lane keeping and overtaking assisting.

A road scene sectioning framework was adopted by Alvarez et al. using a CNN-based algorithm to retrieve the 3D scene layout of the street image from noisy labels combining online and offline learning [79]. The proposed method built with color plane fusion and CNN was able to achieve 95.5% to extract a single image of a lane without manual labelling. This CNN-based approach could be considered as the most efficient method for deploying in unknown environments for AVS road-feature extraction.

However, for each pixel of the image, which was a path or a lane, authors Dong et al. considered the visual road-detection challenge applying a U-Net-prior network with the DAM (Domain Adaptation Model) to reduce the disparity between the training images and the test image [80]. The proposed model was compared to other state-of-art methods such as RBNet [191], StixeNet II and MultiNet [192], where the max-F measures were 94.97%, 94.88% and 94.88%, respectively, in 0.18 s, 1.2 s and 1.7 s. Their methodology obtained 95.57% max F-measurement in 0.15 s faster and more accurately than others, which indicates that their monocular-vision-based systems achieve high precision for a lower running time.

Another kind of approach for storing processes of previous stages, a method based on a combination of CNN and recurrent neural network (RNN), was proposed by Li et al., which was able to identify lane markers using geometry feedback with maximum 99% AUC value [81]. However, since no image pre-processing was conducted, this process took a lot of time in sorting unrelated image areas. In addition, these methods were either time-consuming or inefficient in a true, dynamic world, which does not fulfil the maximum efficiency restriction of a critical function.

The Bayesian method for estimating multihyperbola parameters splitting frames in multiple patches was demonstrated by Fakhfakh et al. to recognize curved lanes under difficult conditions using [82]. The lane line was represented on each component by a hyperbola which was determined using the proposed Bayesian hierarchical model with an average of 91.83% true positive rate (TPR) on the ROMA dataset. To sum up the theory, it could be made more practical by adopting sampling techniques such as Hamiltonian schemes to enhance the model's performance.

Yang et al. suggested a substitution of image pre-processing to reduce the uncertainty about lane state [83]. Their approach uses profound lane detection based on deep learning as a substitute for practical lane detection with UNet encoder including high-grade GPU processing. The paper also states that the CNN-based UNet with Progressive Probabilistic Hough Transformation, UNet, Kalman filter were far more inefficient in terms of identification than the feature-based approaches, such as Hough Transformation (HOG) for lane tracking in real time [84–86].

For predicting lane line under the most challenging conditions, a spatiotemporal-based hybrid architecture after encoding–decoding SCNN and ConvLSTM [87]. This is the very first approach which improves temporal correlation with spatial relation of feature extraction with 98.19% accuracy and 91.8% F1 score. However, although this is one of the strongest approaches, the authors did not apply it to complex weather and nighttime scenarios. Furthermore, to resolve instance level and complex fork and dense line-detection issue, a novel approach was implemented, CondLaneNet, using recurrent instance module applied to a CULane dataset [88]. The approach obtained an 86.10% F1 score while detecting curve lane in complex scenarios despite the lack of proper refining of contextual features.

Multiple deep learning methods were studied regarding the lane curve tracking system. For instance, Dorj et al. deployed circle equation models and parabola equations to redesign the Kalman filter for curved lane tracking with a view to calculating curving parameters in far field view [89]. Although the algorithm had an independent threshold mechanism to compensate for various light conditions, such as low light, further research was needed to identify lane reflections and shadows. The limitation of Dorj et al. was solved in [90], where the authors applied a local adaptive threshold and RANSAC feedback algorithm to prevent misdetection of the lane by estimating two-lane parameter-based issues. Nevertheless, the algorithm did not allow a close-loop lane to maintain lane control

while following the road lane, showing a slightly higher false positive rate (FPR) and slow execution for processing in the CPU only. However, it achieved 99.9% precision, 98.9% accuracy and 99.4% F-measurement in 0.677 fps complex visual and lighting conditions.

Similarly, for overcoming lane detection in complex shadow and lighting conditions full of obstacles, a CNN-based method was presented by Wang et al. [91]. From an inverse perspective, the application of a fixed transformation matrix generated errors as changes occurred, allowing the predicted exhaust point to infinitely shift upward or downward. The authors trained a neural network with a custom loss function that predicted the transformable matrix parameter valued dynamically. The method was implemented on the TuSimple dataset and obtained high accuracy for insufficient light, shadow, missing lane and normal road compared to other deep learning methods, such as Spatial CNN, CNN-FCN and UNet.

As an approach to preventing lighting condition problems for lane detection and tracking, a novel CNN-based model was proposed by Ye et al. [92]. In the pre-processing stage they adopted Yam angle prediction and filtering, followed by segmenting ROIs implementing waveform generation that generated on average 99.25% accuracy in the BIT dataset considering nine cases where day and nighttime accuracies were 99.34% and 98.66%, respectively, as well as a 1.78% average error rate for the Caltech dataset. However, this methodology could be the most suitable candidate only if it is performed with similar outcomes in real-life experiments of level 4 or 5 AVS.

A similar CNN-based approach combined with CNN-LSTM, SegNet and UNet was applied by Zou et al. for lane detection from occlusion scenarios [93]. The method obtained 96.78% accuracy for SegNet and 96.46% for UNet within 46 ms, which was much faster than the average of the other methods. With faster processing and high accuracy, this approach could be considered as one of the most acceptable methods for AVS lane detection.

Jhon et al. proposed a lane-detection algorithm, which calculated the semantic road lane by using the extra tree-based decision forest and DNN from a street scene where hue, saturation, depth (HSD) combined with a deconvolutional network were fine-tuned [94]. In the final stage, a separate extra tree regressor was trained within each lane applying the depths and the manually annotated lane marker locations on the image. The methodology was applied to the TTI and TMD datasets, where it achieved 98.80% and 97.45% accuracy, respectively, for lane detection.

Further, encoder–decoder dilated convolution and finely tuned improvements were implemented by Chen et al. to create a modified CNN road lane detection system called Lane Mark Detector (LMD) which increased the accuracy of the CamVid dataset to 65.2%, obtained 79.6% class average accuracy and increased the test speed to 34.4 fps as well as improved the inference time (29.1 ms) and smaller model size of 66 mb [95].

Moreover, Ghafoorian et al. used Embedding Loss-Driven Generative Adversarial Networks (EL-GAN) for detecting road lanes [96]. This led to even more secure training with stronger discrimination and stabilized the mechanism of adverse preparation. This significantly stabilized the process of opposing training. EL-GAN was also applied to the TuSimple dataset and achieved 96.39% accuracy despite requiring the tuning of a suitable number of parameters. As the loss of embedding into classification boosted the maximum efficiency of the lane marking method, it was one of the best and most appropriate approaches for continuous lane detection and tracking.

Tracking lane during nighttime was one of the most difficult tasks of AVS. He et al. solved the issue by developing a vision-based lane detection system, where they pre-processed with a Gabor filter, continuing adaptive splay ROI and Hough transformation to detect the lane marker [97]. Despite lacking an appropriate self-switching system for defining lanes in all circumstances in pre-processing, the detection rates were 97.31% and 98.15% using two clips of Guangzhou where frame numbers were 3274 and 2231. However, the method faced difficulties when tackling bright light reflection, missing lane marks and lane cracks as well.

Neven et al. formulated a solution using LaneNet and HNet for the problem of lane detection with an instance segmentation problem in which each lane constituted its own instance to be end-to-end trained [98]. In addition, to a set "bird's-eye view", they introduced a learning transfer to the perspective, which was contingent on the image and achieved 96.4% accuracy within 50 fps (frames per second) for the TuSimple dataset. The method was robust enough to adjust the pitch of the ground plane by adapting the transition parameters accordingly, which was the main reason for accurate visualization, and detected lane and lane curves.

Moreover, Kim et al. proposed a fast-learning environment using extreme learning CNN (EL-CNN) combining extreme learning machine (ELM) calculating weights among output and hidden layers in one iteration with CNN for lane marking extraction in complex scenarios to overcome computing of large dataset [99]. It reduced training time 1/50 for the KNU dataset, and 1/200 for the Caltech dataset compared to CNN. Experimental results demonstrate that it obtained maximum weights effectively while maintaining performance of 98.9% accuracy applied to the Caltech dataset.

In another work, Van Gansbeke et al. implemented ERFNet with differentiable least-squares fitting (DLSF) for end-to-end lane detection [100]. The approach used dynamic backpropagation to perform an experiment on a lane detection task that demonstrated that, despite the poor supervision signal, the end-to-end approach exceeded a two-step procedure, scoring 95.80% accuracy in 70 fps applied to the TuSimple dataset. The accuracy was not the maximum, but the weight map did not require post processing for accurate lane estimation.

Hou et al. proposed a lane detection CNN by self-attention distillation (SAD) which had self-learning ability in the training phase and boosted the visual attention of multiple layers in different networks and increased the efficiency of narrow-lane detection systems [101]. The method obtained 96.64% accuracy in the CULane, BDD100 K and TuSimple datasets, although the hyperparameter adjustment was complicated by an insufficient training process and loss functions.

In another work, Liu et al. used a lightweight YOLO network for lane curve detection and tracking with 90.32% precision and 83.76% recall in 50.6 ms [44]. The method was applied to a custom dataset which was evaluated for day and night scenarios. However, the efficiency could be better suited to proper AVS if it solved the interruption of vehicles during lane detection.

In conclusion, most of the approaches have performed well enough to be adopted for practical implementation of AVS. However, modified CNN [92] was able to detect lanes with highest accuracy for both day and nighttime, and the CNN-LSTM-based SegNet and UNet combined approach was [93] able to segment roads within the fastest runtime. The analysis presented some advantages of deep learning methods for lane and road curve detection, for instance, training without manual labelling, reducing computational complexing while in a single frame, lane detection where markers were not clear, in sharp turns and even challenging weather and shadow or low-light conditions. On the other hand, some methods showed huge dependency on dataset pre-labelling, which was inefficient in the long field of view, resource hunger and even not being evaluated in urban traffic scenarios or challenging road conditions. An overview of the deep learning methods reviewed for the detection of lane and road curves is shown in Table 8.

### 3.1.5. Traffic Scene Analysis

Driving scene and driving behavior analysis of autonomous vehicle systems were denoted as the understanding and classifying of driving environment and traffic scene. To discuss the contribution of deep learning to understanding and analyzing complex traffic scenes, several studies were conducted.

**Table 8.** Summary of multiple deep learning methods for lane detection and tracking.

| Ref. | Method | Outcomes | Advantages | Limitations |
|---|---|---|---|---|
| [79] | Modified CNN | 95.5% accuracy for single frame. | Reduced dependency of manual labelling and processing time in single frame. | Deployment for testing did not consider urban or crowded scenarios. |
| [80] | U-Net | 95.57% max F-measurement in 0.15 s. | Smooth segmentation of road surface with multiple objects as obstacles. | Huge dependency on manual pre-labelling. |
| [81] | Multitask CNN and RNN | Max AUC value of 99%. | Recognized region in complex traffic and visualized spatially distributed cues. | Higher computational cost and inefficient for large field of view. |
| [82] | Bayesian Model | 91.83% true positive rate. | Automated curve detection in rural and challenging roads with lower error rate. | Lighting conditions were not considered and slow processing. |
| [83] | UNet and Kalman Filter | 2.5% and 9.75% lateral error generated in 10 ms. | Obtained less lateral error and overcame slow feature extraction. | Limited to a simple close-loop circle in TORCS simulator. |
| [88] | CondLaneNet | 86.10% F1 score. | Solved the lane detection in fork and dense scenarios. | Contextual features need to be refined. |
| [90] | RANSAC | 99.9% precision and 99.4% F-measurement. | Prevent misdetection by estimating parameters when illumination changes. | Slower execution, slightly high FPR and did not consider urban traffic road. |
| [91] | CNN | Obtained highest accuracy (97.85%). | Outperformed in shadow and roads with obstacles. | Created errors during shifting the ground to predict disappearing point |
| [92] | Modified CNN | Average 99.25% accuracy. | Most suitable deployment in two datasets with faster runtime for nine classes | Did not test in real-time driving senses. |
| [93] | CNN-LSTM with SegNet | 96.78% accuracy in 46 ms. | Raised performance in occlusion scenarios. | Combined method of CNN and RNN were resource hungry and slow. |
| [94] | Modified Decision forest and DNN | 98.80% and 97.45% accuracy, respectively. | High accuracy in surface and road detection. | High computational cost. |
| [95] | CNN | 65.2% mIoU, 79.6% class average accuracy. | Modified CNN to achieve low complexity and maintained similar accuracy. | Was not tested in crowded traffic environment. |
| [96] | EL-GAN | 96.39% accuracy. | The loss of embedding in detector boosted the performance closest to the mark. | Required tuning of huge number of parameters. |
| [98] | LaneNet and HNet | 96.4% accuracy within 19 ms. | Did not require post processing, pixel-wise segmentation or fix lane number. | Faced challenges in long field of view while detecting curves. |
| [99] | EL-CNN | 98.9% accuracy. | Reduced training time 1/50 and 1/200 in KNU and Caltech datasets, respectively. | Required matrix inversion for better execution time in high dimensional data. |
| [100] | ERFNet-DLSF | 95.80% accuracy. | Did not need post processing to estimate line coordinates using weight map. | Was not tested in urban or complex lighting conditions. |
| [101] | CNN with SAD | 96.64% accuracy. | Has self-learning ability and increased the efficiency of narrow-lane detection. | Complex hyperparameter adjustment for inadequate training process. |
| [44] | Lightweight YOLO Network | 90.32% precision and 83.76% recall in 50.6 ms. | Applicable in both day and night scenarios and multiple fields of view. | High interruption for obscured vehicles. |

To contribute to this field for developing traffic scene analysis for AVS, Geiger et al. proposed a novel method of generative probabilism to understand traffic scenes with the Markov Chain Monte Carlo, which was used to deal with the dynamic relationship between crossroads and feature presentation [102]. The human-inspired method took the benefit from a wide range of visual cues through the form of vehicle directions, vanishing points, semantic scene labels, scenario flow and grids rather than requiring sensor values

such as LiDAR and GPS, where most of the standard methods struggled for most of the intersections due to the lack of these attribute labels. Moreover, the method can accurately identify urban intersections with up to 90% accuracy at 113 real-world intersections.

Another scene semantic segmentation approach is the High-Resolution Network (HRNet) proposed by Wang et al. [103], where the method obtained 81.1% mIoU. HRNet linked the high-to-low resolution convolution streams in parallel and transferred data across repeatedly. The advantage of the method was that the resulting representation was richer semantically and spatially. However, it required huge memory size due to high resolution-wise segmentation. Additionally, the same author improved their previous work applying contrastive loss to previous architecture (HRNet), which explored pairwise pixel-to-pixel dependencies applied to the Cityscape dataset and obtained 1.1% higher mIoU [104]. Although the proposed method demonstrated effective performance, which is applicable for top-tier AVS, it was unable to achieve success during contrastive learning in few parts of the labelled dataset. To tackle this issue, Zhao et al. [105] presented a contrastive approach following previous research [103,104] and proposing SoftMax tuning rather than applying contrastive loss and cross-entropy at once. The authors demonstrated three variants of label and pixel-wise contrastive losses by adopting DeepLabV3 with ResNet-50 with 256 channels of convolution layers and bilinear resizing for input resolution for semantic segmentation. This approach showed 79% and 74.6 mIoU, respectively, for Cityscape and PASCAL VOC 2012 datasets but using 50% less labelled dataset. Thus, powerful semantic segmentation with a fine-tuned pretrained method can be a major pathway for higher level AVS for scene analysis.

Furthermore, to develop a scene recognition framework, Tang et al. demonstrated GoogleNet for multi-stage feature fusion, named G-MS2F, segmented into three layers to feature extractions and scoring scene understanding, that can be efficiently employed for autonomous driving systems [106]. The framework obtained 92.90%, 79.63% and 64.06% accuracy, respectively, when applied to the Scenel5, MIT67 and SUN397 datasets for image scene recognition.

Similarly, a multiresolution convolutional neural network architecture was proposed by Wang et al. for driving scene understanding in different scales where they used two categories of resolution images in the input layer [107]. A combination of fine-resolution CNN and coarse-resolution CNN was included for recording small and comparatively large-scale visual frameworks. To obtain visual information with more accurate resolution and enhanced spatial information, on an inception layer, three convolutional layers were added. They implemented the architecture on the Place365 dataset where the lowest error rate was 13.2%.

Moreover, a 2D-LSTM model was proposed to learn information from surrounding context data of scene labels as well as spatial dependencies in [108] within a single model that generated each image's class probabilities. They obtained 78.52% accuracy when deploying on the Standford background dataset.

Fu et al. introduced an integrated channel contextual framework and spatial contextual framework as a contextual deconvolution network (CDN) that used both local and global features [109]. In an attempt to optimize the visualization of the semantic data, the decoder network utilized hierarchical supervision for multilevel feature maps in the Cityscapes dataset and achieved 80.5% mean IoU.

Following the work, an optimized model of a deep neural network was proposed with two distinct output directions by Oeljeklaus et al. Their method foresaw road topology along with pixel-dense categorization of images at the same time, and lower computing costs were offered in real-time autonomous applications via a proposed architecture combined with a novel Hadamard layer with element-wise weights using Caffe and achieved 0.65 F1, 0.67 precision and recall 0.64 after fine-tuning the architecture with 10,000 iterations [110]. Although strong restrictions placed by the double-loss function on the DNN feature maps caused difficulties in optimizing the process, research in relation to the

Cityscapes dataset showed that a sufficient representation of traffic scene understanding was achieved relying on broad traffic components.

In another work, Xue et al. presented a CNN with Overlapping Pyramid Pooling (OPP) applied to sematic segmentation of city traffic area based on a fisheye camera with wider vision [111]. The OPP was demonstrated for the exploratory study of the local, global and pyramidal local context information to resolve the complicated scenario in the fisheye image. Furthermore, they built novel zoom augmentation for augmenting fisheye images to boost performance of the method where it scored 54.5 mIoU, which is higher than the standard OPP-Net and Dilation10 method. This approach could be highly suitable for short FoV traffic scene understanding in urban areas.

Pan et al. proposed Spatial CNN, a CNN-like framework for efficient spatial distribution of information through slice-by-slice message passing from the top hidden layer [112]. It was tested at two roles: lane recognition and traffic scene perception. The analysis showed that the continuity of the long, small structure was appropriately preserved by SCNN, while its diffusion effects have proven positive for large objects in semantic segmentation. However, SCNN can master the spatial relationship for the structural production and increase operating efficiency, showing that SCNN was 8.7% and 4.6% superior to the recurrent neural network (RNN) focused on ReNet and MRF + CNN (MRFNet). It scored 68.2 mIoU for semantic segmentation and achieved 96.53% on the TuSimple Benchmark Lane Detection Challenge combined with traffic scene analysis.

Mou et al. proposed a vision-based vehicle behavior prediction system by incorporating vehicle behavior structural information into the learning process, obtaining a discrete numerical label from the detected vehicle [113]. The OPDNN (overfitting-preventing DNN) was constructed using the structured label as final prediction architecture, and after more than 7000 iterations, 44.18% more accuracy on-road vehicle action than CNN was achieved. In addition, the method decreased the issue of overfitting in a small-scale training set and was highly efficient for analysis of on-road vehicle behavior predicting turning angles.

In another work, Jeon et al. proposed a model built on the CNN and Long Short-Term Memory (LSTM) networks to predict risk of accidents and minimize accidents and analyzing traffic scenes differing conditions of driving such as lane merging, tollgate and unsigned intersections [114]. They implemented a multi-channel occupancy Grid Map (OGM) as a bird's-eye view that ostensibly included the features of many interaction groups to represent the traffic scene [85].

Additionally, the CNN was used to derive numerous inter-vehicle interactions from the grid and to estimate possible time-serial predictions of the derived functions. For instance, Lui et al. demonstrated a deep understanding of the vehicle-specific scene understanding state-of-art in terms of using traffic environment as object joining automatic scene segmentation and object detection, which reduced the person manipulation [55]. A SegNet network whose weight was initialized by the VGG19 network was used for semantic segmentation on the Auckland traffic dataset [115].

Afterwards, a Faster RCNN-based approach transformed feature maps in the ROI (ROI) and transferred those to the classification mode. It had an accuracy of 91% for sky detect, 90% for bus lane, 86% for road, 70% for lane and 81% building classes applying VGG19-SegNet. However, it suffered from false rate for not having a high-resolution labelled dataset and a weak vehicle detection process.

Furthermore, two state-of-the-art versions of machine learning and deep learning (DNN) were used by Theofilatos et al. to estimate the incidence of a crash in real time where the dataset comprised historical accident information and combined current traffic and weather information from Attica Tollway, Greece [116]. The method achieved accuracy, precision, recall and AUC of 68.95%, 52.1%, 77% and 64.1%, respectively. The limitation was the transferability while returning the parameters and the absence of good interplay during comparison and insufficiently clarified unexpected heterogeneity. The possible solution offered by the authors was to apply a sensitivity analysis that was not used when applying a binary logistic model in their work to determine risk of crashes.

Moreover, Huegle et al. proposed a Graph-Q and DeepScene-Q off-policy reinforcement learning-based approach for traffic scene analysis and understanding applied to a custom dataset [117]. The proposed method used dynamic awareness-based scene understanding for AVS, although it was not tested in a real driving environment and was unable to track lanes while moving quickly.

With a view to understanding hazardous or damaged roads in a driving situation for a smooth autonomous driving experience, deep learning approaches can also provide a solution. Nguyen et al. used CNN architecture to identify damages and cracks in the road that reduced false detection and without pre-processing, which helped to decrease computational time [118]. On the other hand, authors adopted a principal component analysis (PCA) method and CNN to classify and sense damaged roads with their own dataset.

Another Deep CNN-based approach with discriminative features for understanding road crack identification was developed by Zhang et al., which also could be a pathway to implement in AVS [119]. The core advantage of the framework was self-learning features that did not rely on manual labelling and geometrical pavement predictions. An alternative method for autonomous road cracks alongside pothole detection was demonstrated by Anand et al. as part of an analysis of traffic scene [120]. SegNet was applied with texture that relied on features to separate roads from traffic scene in order to build a first division of mask and concatenated with the second masking, which was created with a 2-canny edge algorithm and dilation. Further, SqueezeNet was applied to the GAPs dataset along with being prepared for deployment in a self-driving vehicle. Compared with a similar approach of Zhang [119], it achieved higher precision, recall and F1 score, leaving one drawback where it failed to recognize cracked road that was misinterpreted as under-construction surface texture. For this outcome, the method of Anand et al. [120] was a more suitable approach for identifying damage road surface.

In summary, deep learning approaches such as fine-resolution CNN and coarse-resolution CNN, 2D-LSTM model RNN, HRNet, Deep CNN, Contextual Deconvolution Network, DNN and CNN with pyramid pooling were analyzed which demonstrated high-accuracy traffic-scene understanding from a crowded movable platform, showed less model complexity, being applicable in different scales, avoiding confusion of ambiguous labels by increasing the contrast among pixels, in some cases developing more expressive spatial features and predicting risk of accident. However, some approaches were limited for implementation because of the requirement of re-weighting, which was inapplicable in uncertain environments, slower computational time, low accuracy and inability to focus on objects in dim light and foggy vision. The overall summary is presented in Table 9.

### 3.2. Decision Making

As the world economy and technology have grown and developed, vehicular ownership has increased rapidly, along with over one million traffic incidents worldwide per year. Statistics indicate that 89.8% of incidents took place because of wrong driver decision-making [193]. To solve this issue with the concept of AVS, the decision-making process was one of the key fields for studying a combined deep learning and deep reinforcement learning-based approach to take humanlike driving decisions when accelerating and decelerating, lane shifting, overtaking and emergency braking, collision avoidance, vehicle behavior analysis and safety assessment.

For instance, the automated driving coordination problem was defined as a problem of the Markov Decision Process (MDP) in the research of Yu et al., during the simulation of vehicle interactions applying multi-agent reinforcement learning (MARL) with a dynamic coordination graph to follow lead vehicles or overtaking in certain driving scenarios [121]. The advantage of the method was when most of the study focused on single vehicle policy, the proposed mechanism resolved the limitation of coordination problem in autonomous driving during overtaking and lane-shifting maneuvers, obtaining higher rewards than rule-based approaches.

**Table 9.** Summary of multiple deep learning methods for traffic scene analysis.

| Ref. | Method | Outcomes | Advantages | Limitations |
|---|---|---|---|---|
| [55] | VGG-19 SegNet | Highest 91% classification accuracy. | Efficient in specified scene understanding, reducing the person manipulation. | Showed false rate for not having high-resolution labelled dataset. |
| [102] | Markov Chain Monte Carlo | Identify intersections with 90% accuracy. | Identified intersections from challenging and crowded urban scenario. | Independent tractlets caused unpredictable collision in complex scenarios. |
| [103] | HRNet | 81.1% mIoU. | Able to perform semantic segmentation with high resolution. | Required huge memory size. |
| [104] | HRNet + contrastive loss | 82.2% mIoU. | Contrastive loss with pixel-to-pixel dependencies enhanced performance. | Did not show success of contrastive learning in limited data-labelled cases. |
| [105] | DeepLabV3 and ResNet-50 | 79% mIoU with 50% less labelled dataset. | Reduce dependency on huge labelled data with softmax fine-tuning. | Dependency on labelled dataset. |
| [106] | Multistage Deep CNN | Highest 92.90% accuracy. | Less model complexity and three times less time complexity than GoogleNet. | Did not demonstrate for challenging scenes. |
| [107] | Fine- and coarse-resolution CNN | 13.2% error rate. | Applicable at different scale. | Multilabel classification from scene was missing. |
| [108] | 2D-LSTM with RNN | 78.52% accuracy. | Able to avoid the confusion of ambiguous labels by increasing the contrast. | Suffered scene segmentation in foggy vision. |
| [109] | CDN | Achieved 80.5% mean IoU. | Fixed image semantic information and outperformed expressive spatial feature. | Unable to focus on each object in low-resolution images. |
| [110] | DNN with Hadamard layer | 0.65 F1 score, 0.67 precision and 0.64 recall. | Foresaw road topology with pixel-dense categorization and less computing cost. | Restrictions by the double-loss function caused difficulties in optimizing the process. |
| [111] | CNN with pyramid pooling | Scored 54.5 mIoU. | Developed novel image augmentation technique from fisheye images. | Not applicable for far field of view. |
| [112] | Spatial CNN | 96.53% accuracy and 68.2% mIoU. | Re-architected CNN for long continuous road and traffic scenarios. | Performance dropped significantly during low-light and rainy scenarios. |
| [113] | OP-DNN | 91.1% accuracy after 7000 iterations. | Decreased the issue of overfitting in small-scale training set. | Required re-weighting for improved result but inapplicable in uncertain environment. |
| [114] | CNN and LSTM | 90% accuracy in 3 s. | Predict risk of accidents lane merging, tollgate and unsigned intersections. | Slower computational time and tested in similar kinds of traffic scenes. |
| [116] | DNN | 68.95% accuracy and 77% recall. | Determined risk of class from traffic scene. | Sensitivity analysis was not used for crack detection. |
| [117] | Graph-Q and DeepScene-Q | Obtained *p*-value of 0.0011. | Developed dynamic interaction-aware-based scene understanding for AVS. | Unable to see fast lane result and slow performance of agent. |
| [118] | PCA with CNN | High accuracy for transverse classification. | Identified damages and cracks in the road, without pre-processing. | Required manual labelling which was time consuming. |
| [119] | CNN | 92.51%, 89.65% recall and F1 score, respectively. | Automatic learning feature and tested in complex background. | Had not performed in real-time driving environment. |
| [120] | SegNet and SqueezedNet | Highest accuracy (98.93%) in GAPs dataset. | Identified potholes with texture-reliant approach. | Failed cases due to confusing with texture of the restoration patches. |

In another work, the Driving Decision-Making Mechanism (DDM) was built by Zhang et al., using an SVM algorithm, optimized with the weighted hybrid kernel function and a Particle Swarm Optimization algorithm to solve decision-making issues including free driving, tracking car and lane changing [122]. The proposed decision-making mechanism obtained 92% accuracy optimizing an SVM model compared with RBF kernel and BPNN model, where the evaluated performance shows that free driving achieved 93.1% and tracking car and lane changing achieved 94.7% and 89.1% accuracy, respectively, in different

traffic environments within 4 ms for average reasoning time. The authors presented a hypothesis when analyzing the results: for driving decisions, road conditions have nearly no effect on heavy traffic density. Despite achieving good accuracy, some limitations were mentioned, such as not applying to real-world driving environments and not yet investigating critical driving scenes such as sudden presence of pedestrians or objects.

This issue of [122], was solved by Fu et al., who proposed autonomous braking, analyzing a lane-changing behavior decision-making system for emergency situations, implementing an actor-critic-based DRL (AC-DRL) with deep deterministic policy gradient (DDPG) and setting up a multi-object reward function [123,124], obtaining 1.43% collision rate. The authors mentioned that using a large training dataset online can be tough and expensive, and the continuous action function decreased the convergence rate and can quickly be lowered to the maximum local.

Moreover, to overcome the limitation of reinforcement learning in complex urban areas, Chen et al. used model-free deep reinforcement learning approaches named Double Deep Q-Network (DDQN), Twin Delayed Deep Deterministic Policy Gradient (TD3) and Soft Actor-Critic (SAC) to obtain low dimensional latent states with visual encoding [125]. They improved performance by implementing a CARLA simulator by altering frame dropping, exploring strategies and using a modified reward and network design. The method was evaluated in one of the most complicated tasks, a busy roundabout, and obtained improved performance compared to baseline. In the 50 min test, the three approaches were able to enter with high success rate but performance of DDQN and TD3 decreased after covering a long distance. In the best case, SAC achieved 86%, 80%, 74%, 64%, 58% success rate for first, second, third, desired exits and goal point, respectively, where DDQN and TD3 had an almost zero success rate for desired exit and goal point arriving.

To avoid training complexity in a simulation environment, the DDPG algorithm with actor-critic method was applied in [124] using deep reinforcement learning (DRL), considering three reward function braking scenarios: braking too early and too late, and too-quick braking deceleration. The outcomes of their proposed methodology showed that the error collision rate was 1.43% which was gained by evaluating the performance of the diverse initial positions and initial speed strategies. The ratio of obtaining maximum deceleration was 5.98% and exceeding jerk was 9.21%, which were much improved compared to DDPG with steering and DQN with discrete deceleration.

A dueling deep Q-network approach was demonstrated by Liao et al. to make a strategy of highway decision making [126]. The method was built for lane-changing decisions to make a strategy for AVS on highways where the lateral and longitudinal motions of the host and surrounding vehicles were manipulated by a hierarchical control system. The outcomes showed that after 1300, 1700, 1950 episodes, the approach was able to avoid collision after 6 h of training and 26.56 s of testing.

In another study, Hoel et al. introduced a tactical framework for a decision-making process of AVS combining planning with a DRL-extended Alpha Go algorithm [127]. The planning phase was carried out with a modification in the Monte Carlo Tree Search, which builds a random sampling search tree and obtained a 70% success rate in highway cases. The contrast between traditional MCTS and the variant in this search was that a neural network formed through DRL aimed towards the search tree's most major aspects and decreased the essential sample size and helped to identify long temporal correlations with the MCTS portion. However, the proposed process considered 20 simulation parameters and 11 inputs to a neural network which were very efficient and made more suitable for practical implementation.

Overtaking maneuvers for intelligent decision making while applying a mixed observable Markov decision process was introduced by Sezer, solving overtaking maneuvers on two-track roads [128]. In this paper, the author presented a new formulation for the issue of double-way overtaking by the resources of the mixed observability MDP (MOMDP) to identify the best strategy considering uncertainties. This was used for overcoming the problem, and was illustrated by the active solvers' growth and in cognitive technological

advances by reducing time-to-collision (TTC) methods in different simulations. The method surpassed nine periods, relative to both MDP and conventional TTC methods. However, the limitation of proper discretion can also be considered with respect to the actual speed and distance values. A higher number of states that were specifically connected for computing and MOMDP algorithm tend to be required as the actual implementation hindrance.

To overcome the issue of vehicle overtaking which needs an agent to resolve several requirements in a wide variety of ways, a multigoal reinforcement learning (MGRL)-based framework was introduced to tackle this issue by Ngai et al. [129]. A good range of cases of overtaking were simulated to demonstrate the feasibility of the suggested approach. When evaluating seven different targets, either Q-Learning or Double Action QL was being used with a fusion function to assess individual decisions depending upon the interaction of the other vehicle with the agent. The hypothesis of the work was that this proposal was very efficient at taking accurate decisions while overtaking, collision avoiding, arriving on target timely, maintaining steady speed and steering angle.

Brännström et al. presented a collision-avoiding decision-making system adopting a Bayesian network-based probabilistic framework [130]. A driver model enabled the developer to carry out early actions in many circumstances in which the driver finds it impossible to anticipate the potential direction of other road users. Furthermore, both calculation and prediction uncertainties were formally discussed in the theoretical framework, both when evaluating driver adoption of an action and when predicting whether the decision-making method could avoid collision.

Another important decision-making task is intelligent vehicle lane-changing policy. Based on the area of acceleration and braking mechanism, a method was introduced by Zhu et al. [131]. First, velocity and relative distance acceleration area was developed based on a braking mechanism and acceleration was used as a safety assessment predictor and then, a method for lane changing with the accelerating field was built, while the driver's behaviors, performance and safety were taken into consideration. In compliance with the simulation findings, the use of lane-changing decision-making strategies based on the acceleration can be optimized with driver behaviors for lane-change steps, including starting line, span and speed establishing safety at the same time.

Although previous approaches presented a decision-making mechanism for lane changing, most of them did not show DMS for behavior prediction while lane changing [132]. A fuzzy interface system with an LSTM-based method for AVS was proposed by Wang et al. to analyze behavior of surrounding vehicles to ensure safety while lane changing with 92.40% accuracy. The novelty of their work was the adjustment of motion state dynamically in advance.

Li et al. proposed a framework for the analysis of the behavior, using a gradient-boosting decision tree (GBDT), merging acceleration or deceleration behavior with the data from the trajectory of the vehicle processed in the noise method on the U.S. highway 101 [133]. The partial dependency plots demonstrated that the effect on the fusion of acceleration or deceleration in independent variables by understanding the key impacts of multiple variables, was non-linear and thus distinct from the car tracking behavior with 0.3517 MAD (Mean Absolute Deviation) value, which suggested that the adoption of typical vehicle models in combination results cannot reflect characteristic behavior.

Further, DRL with Q-masking was applied by Mukadam et al. to make tactical decisions for shifting lanes [134]. They introduced a system which provided a more organized and data-efficient alternative to a comprehensive policy learning on issues where high-level policies are difficult to formulate through conventional optimization or methods based on laws. The success rate of 91% was 21% higher than human perception and the 0% collision was 24% lower than human perception. This method of DRL with Q-masking worked best in the case of avoiding collision while lane shifting.

Similarly, Wang et al. adopted DRL but combined with rule-based constraints to take lane-changing decisions for AVS in a simulated environment and MDP, which was challenging for high-level policy to develop through conventional methods of optimization

or regulation [135]. The training agent could take the required action in multiple situations due to the environment of state representation, the award feature and the fusion of a high level of lateral decision making and a rule-based longitudinal regulation and trajectory adjustment. The method was able to obtain a 0.8 safety rate with superior average speed and lane-changing time.

Chae et al. demonstrated an emergency braking system applying DQN [136]. The problem of brake control model was conceived in Markov's decision-making process (MDP), where the status was provided by the relative location of the hazard and the speed of the vehicle and the operating space specified as the collection of brake actions including no braking, weak, medium and heavy braking operation, combining vehicle, pedestrian and multiple road conditions scenarios, and the obtained collision rate decreased from 61.29% to 0% for a TTC value from 0.9 s to 1.5 s. As a result, this DQN-based approach was selected as one of the most practical systems for SVM in terms of autonomous braking.

Furthermore, to analyze high-accuracy braking action from a driving situation declaring four variables, that is, speed of host vehicle, time to collision, relative speed and distance between host and lead vehicle, Wang et al. used hidden Markov and Gaussian mixture-based (HMGM) approach [137]. The efficient technique was able to obtain high specificity and 89.41% accuracy despite not considering kinematic characteristics of lead or host vehicle for braking. However, the analysis of four variants while braking could be a pathway to develop an improved version of braking decision making for AVS.

When most of the approaches had dependency on datasets, methods such as DRL that combined DL and RL were extremely efficient for driving decision making in an unknown environment. For example, Chen et al. developed a brain-inspired simulation based on deep recurrent reinforcement Q-learning (DRQL) for self-driving agents with better action and state space inputting only screen pixels [138]. Although the training process was long, it resulted in better-than-human driving ability and Stanford driving agent in terms of reward gain, which indicates that this approach was one of the most suitable for applying in AVS.

Another DRL-based approach combined with automatically generated curriculum (AGC), was extremely efficient for intersection scenarios with less training cost [139]. The method obtained 98.69% and 82.1% mean average reward while intersection approaching and traverse. However, the approach might lack proper finishing or goal researching in some cases of intersection traverse, but it is still very efficient for not depending on pre-trained datasets.

Similarly, continuous decision-making for intersection cases in top three accident-prone crossing paths in a Carla simulator using DDPG and CNN surpassed the limitation of single scenario with discrete behavior outputs fulfilling the criteria for safe AVS [140]. DDQG was utilized to address the MDP problem and find the best driving strategy by mapping the link between traffic photos and vehicle operations through CNN that solved the common drawback of rule-based RL methods deployed in intersection cases. The method obtained standard deviation (SD) values for left turn across path opposite direction and lateral direction, straight crossing path 0.50 m/s, 0.48 m/s and 0.63 m/s, respectively, although it only considered lateral maneuvers and two vehicles in the intersection.

In contrast, approach was introduced by Deshpande et al. for dealing with behavioral decision making for environments full of pedestrians [141]. Deep recurrent Q-network (DRQN) was used for taking safe decisions to reach a goal without collision and succeeded in 70% of cases. With the comparatively lower accuracy, this approach also could be very appropriate if deep learning agents were added for better feature analysis.

For AVS navigation avoiding on-road obstacles, a double deep Q-learning (DDQN) and Faster R-CNN in a stochastic environment obtained stable average reward value after only 120 epochs with maximum 94% accuracy after 180,000 training steps with hyperparameter tuning [142]. However, this approach only considered vehicles in parallel and did not show how DDQN and Faster R-CNN are fused. Moreover, the approach was still unable to obtain stable performance in uncertain moments.

Mo et al. demonstrated reinforcement learning agent and an MCTS-based approach to reduce safe decision making and behaviors by safe policy search and risk state prediction module [143]. This research assessed the challenge of decision making for a two-lane overtaking situation using the proposed safe RL approach and comparing it with MOBIL and DRQN. The proposed model outperformed MOBIL and DRQN by scoring 24.7% and 14.3% higher overtaking rate with 100% collision-free episodes and highest speed. Therefore, the proposed Safe RL could be a pathway for current AVS for risk-free trajectory decision making.

In conclusion, decision making is the most vital part of an intelligent system, and to obtain acceptable human-like driving decisions, multiple deep learning and deep reinforcement learning methods were analyzed (shown in Table 10). The discussed approaches where able to resolve severe limitations and outperformed in overtaking, braking, behavioral analysis and significant segments of decision making for full AVS.

**Table 10.** Summary of multiple deep learning methods for decision-making process.

| Ref. | Method | Outcomes | Advantages | Limitations |
|------|--------|----------|------------|-------------|
| [121] | MARL | Obtained higher rewards than expert rule-based approach. | Resolved the limitation of coordination problem in autonomous driving. | Individual learning of RL agents involved high computational cost. |
| [122] | Weighted hybrid SVM | Max 94.7% accuracy for lane changing task. | Faster decision making in different traffic conditions. | Yet to demonstrate for critical and uncertain driving scenes. |
| [131,132] | AC-DRL with DDPG | 1.43% collision rate. | Autonomous braking system while lane shifting with high accuracy. | Complexity with large training dataset and decreased the convergence rate. |
| [125] | DDQN, TD3, SAC | In best case SAC achieved 86% success rate. | Decreased sample complexity with visual encoding. | Lack of exploration caused failure cases of DDQN and TD3. |
| [126] | Dueling DQN | Able to avoid collision after lowest 1300 episodes. | Develop lane-changing decision-making strategy for AVS on highway. | Still needed to improve training process for feasible decision making strategy. |
| [127] | Monte Carlo Tree Search | 70% success rate for highway exist case. | Combines planning stage to make efficient driving decision on highway. | Required huge training samples and did not consider lead vehicles' behavior. |
| [128] | MOMD + SARSOP | 91.67% les collision and 25.9% enhanced response rate. | Efficient overtaking decision without rule-based system and optimum actions. | Did not consider real-time speed and distance value. |
| [129] | MGRL | Almost 100% safety index to reach goal without collision. | Outperformed overtaking, collision avoiding, arriving at seven RL goals. | Insignificant performance to keep lane while overtaking. |
| [130] | Bayesian network | Higher driver acceptance while avoiding pedestrians. | Collision avoidance with path prediction and threat assignation. | Testing in real and challenging driving scene was not mentioned. |
| [131] | Polynomial trajectory | Better lateral and steering angle than ground truth. | Able to perform emergency braking decision at safe distance. | Only performed best in straight-road scenario. |
| [132] | LSTM with Fuzzy Logic | 92.40% accuracy. | Decision based on the behavior of surrounding vehicles. | Urban or real-life traffic conditions were not considered. |

**Table 10.** *Cont.*

| Ref. | Method | Outcomes | Advantages | Limitations |
|------|--------|----------|------------|-------------|
| [133] | GBDT | Calibration approach scored 0.3517 MAD. | Understand the key impacts of multiple variables on acceleration of fusion. | Implemented on old dataset and driver features were not analyzed. |
| [134] | DRL using Q-masking | 91% success rate with 24% lower collision rate. | Effective on high-level policies learning through conventional optimization. | Did not analyze real-time and complex road challenges. |
| [135] | Rule-based DQN policy | Safety rate 0.8 on average speed and lane-changing time. | Productive data alternative to end-to-end policy learning on challenge for high-level policy. | Required explicit driving path while training that caused low performance in complex scenes. |
| [136] | DQN | Collision rate decreased from 61.29% to 0%. | Efficient and quick emergency braking in complex environment. | Ambiguous training and environment setting. |
| [137] | HMGRM | Achieved specificity 97.41% and 89.41% accuracy. | Highly accurate braking action from driving situation. | Analysis of kinematic characteristics for both host and lead vehicle was missing. |
| [138] | DRQN | Obtained maximum 64.48% more reward than human. | Did not require a labelled dataset and only took screen pixel as input. | Training time consuming. |
| [139] | AGC based DRL | 98.69% higher average mean reward at intersection. | Efficient on intersection scenarios and reduced DRL training time. | Showed few collisions and unfinished cases during intersection traverse. |
| [140] | CNN + DDPG | Lowest 0.48 SD for left turn across path. | Overcame drawback of single scenario with discrete behavior in intersections. | Only considered lateral maneuvers. |
| [141] | DRQN | 70% success rate for collision free episodes. | Tackled high-level behavioral decision in pedestrian-filled environment. | Low accuracy. |
| [142] | DDQL and FRC | Maximum 94% accuracy with stable reward. | Applied for four driving decision for navigation avoiding obstacle. | Limited decision making for vehicles in parallel sides. |
| [143] | RL + MCTS | 100% collision free episodes. | Outperformed DRQN and MOBIL method for safe lane shifting | Did not consider urban scenarios. |

### 3.3. End-to-End Controlling and Prediction

End-to-end controlling is one of the major fields of study for AVS. Human mistakes were the main cause of road accidents, and fully autonomous vehicles can help reduce these accidents.

To improve the control system of AVS analyzing driving scenarios for lane changing, An et al. [144] proposed a system that tried to approximate driver's actions based on the data obtained from an uncertain environment that were used as parameters while transferring to parameterized stochastic bird statecharts (stohChart(p)) in order to describe the interactions of agents with multiple machine learning algorithms. Following that, a mapping approach was presented to convert stohChart(p) to networks of probabilistic timed automata (NPTA) and this statistical model was built to verify quantitative properties [145]. In the learning case, weighted KNN achieved highest accuracy combined with the proposed method considering training speed and accuracy, where it achieved 85.5% accuracy in 0.223 s and in the best case, time cost for probability distribution time for aggressive, conservative and moderate driving styles was 0.094, 0.793 and 0.113 s, respectively. The authors categorized

their work into learning phase, modelling phase and quantitative analyzing phase in order to develop the driving decision-taking phase.

A method was demonstrated by Pan et al. to control independently at high speeds using human-like imitation learning, involving constant steering and acceleration motions [146]. The dataset's reference policy was derived from a costly high-resolution model predictive controller, which the CNN subsequently trained to emulate using just low-cost camera sensors for observations. The approach was initially validated in ROS Gazebo simulations before being applied to a real-world 30 m-long dirt track using a one-fifth-scale car. The sub-scale vehicle successfully learnt to navigate the track at speeds of up to 7.5 m/s.

Chen et al. focused on a lane-keeping end-to-end learning model predicting steering angle [147]. The authors employed CNN to the current NVIDIA Autonomous Driving Architecture, where both incorporated driving image extraction and asserting steering angle values. To test the steering angle prediction while driving, they considered the difference among ground truth angle which was generated by human drivers vs. predicted angle where they acquired higher steering prediction accuracy with 2.42 mean absolute error and suggested for data augmentation for training to achieve a better performance.

In another work, a technically applied system of multitask learning in order to estimate end-to-end steering angle and speed control, was proposed in [148]. It was counted as one of the major challenging issues for measuring and estimating speed only based on visual perceptions. Throughout their research, the authors projected separation of speed control functions to accelerate or decelerate, using the front-view camera, when the front view was impeded or clear. Nevertheless, it also showed some shortcomings in precision and pre-fixed speed controls. By combining previous feedback speed data as a complement for better and more stable control, they improved the speed control system. This method could be stated to solve error accumulation in fail-case scenarios of driving data. They scored $1.26°$ Mean Absolute Error (MAE) in estimating real-time angles along with 0.19 m/s and 0.45 MAE on both datasets for velocity prediction. Thus, the improved result made the method one of the most applicable versions of CNN and data-driven AV controlling. While driving, people identify the structures and positions of different objects including pedestrians, cars, signs and lanes with human vision. Upon recognizing several objects, people realize the relation between objects and grasp the driving role. In the spatial processing of single images by the application of three-dimensional vectors, CNN has certain shortcoming in the study of time series. However, this issue cannot be overcome using CNN alone.

To solve this limitation Lee et al. demonstrated an end-to-end self-driving control framework combining a CNN and LSTM-based time-series image dataset applied in a Euro Truck simulator [149]. The system created a driving plan which takes the changes into account over time by using the feature map to formulate the next driving plan for the sequence. Moreover, NVIDIA currently has succeeded in training a ConvNet for converting raw camera images into control steering angles [150]. It resolved end-to-end control by predicting steering angle without explicating labels with approximately 90% autonomy value and 98% autonomous of the testing period. This approach was one of the most demonstrated approaches that boosted research of AVS applying deep learning methods.

A similar method, deep ConvNet, was used by Chen et al. to train for directly extracting the identified accessories from the front camera [151]. A basic control system, based on affordance principles, provided steering directions and the decision to overtake proceeding vehicles. Rather than using lane-marking detection methods as well as other objects to assess indirect activity specifications of the car, a variety of driving measures allowances were specified. This method included the vehicle location, the gap to the surrounding lane markers and records of previous car driving. While this was a very trendy concept, for many reasons it may be challenging to handle traffic with complex driving maneuvers and make a human-like autonomous vehicle controlling system.

To deploy a human-like autonomous vehicle speed-control decision-making system Zhang et al. proposed a double Q-network-based approach utilizing naturalistic driving

data built on the roads of Shanghai inputting low dimensional sensor data and high-dimensional image data obtained from video analysis [152]. They combined deep neural networks and double Q-learning (DDQL) [194–196] to construct the deep Q-network (DQN) model which was able to understand and make optimal control decisions in simultaneous environmental and behavioral states. Moreover, real-world data assessment reveals that DDQN can be used on a scale to effectively minimize these unreliable DQN problems, resulting in more consistent and efficient learning. DDQN had increased both in terms of interest precision and policy efficiency. The model performed 271.13% better than DQN in terms of speed-control decision making. Even so, the proposed approach could be more applicable to an unknown driving environment with combined CNN agent for feature extraction.

Chi et al. formulated a ST-LSM network that incorporates spatial and temporal data from previously multiple frames from a camera's front view [153]. Several ST-Conv layers were used in the ST-LSTM model to collect spatial information and a layer of Conv-LSTM was used to store temporarily data at the minimal resolution on the upper layer. However, the spatial and temporal connection among various feature layers was ignored by this end-to-end model. They obtained a benchmarking 0.0637 RMSE value on the Udacity dataset, creating the smallest 0.4802 MB memory and 37.107 MB model weight. The limitation of the paper was that all present end-to-end driving models were only equipped by focusing on the ground truth of the current frame steering angle, which indicated a lack of further spatiotemporal data.

Furthermore, to obtain a better control system, the previous issue was tackled, and an end-to-end steering control system was implemented by Wu et al. by concatenating future spatiotemporal features [154]. They introduced the encoding for an advanced autonomous driving control system of spatiotemporal data on a different scale for steering angle approximation using the Conv-LSTM neural framework with a wide-spectrum spatiotemporal interface module. Sequential data were utilized to improve the space-time expertise of the model during development. This proposed work was compared with end-to-end driving models such as CgNet, NVIDIA's PilotNet [155] and ST-LSTM Network [153], where the root mean square error (RMSE) was 0.1779, 0.1589 and 0.0622, respectively, and showed the lowest RMSE value of 0.0491 to predict steering angles, which was claimed to be more accurate than an expert human driver. Thus, this approach was applicable for a level 4 or 5 autonomous vehicle control system.

Moreover, a deep neural network-based approach with weighted N-version Programming (NVP) was introduced for resilient AV steering controlling [156]. Compared to the other three networks (chauffeur, autumn, rambo), the proposed network showed 40% less RMSE retrieving steering angles in clear, rain, snow, fog and contrast lighting conditions. However, there was a high failure rate for the large developing cost for training an individual DNN model.

Aiming to build a vehicle motion estimation system for diversity awareness while driving, Huang et al., via latent semantic sampling [157], developed a new method to generate practical and complex trajectories for vehicles. First, they expanded to include semantic sampling as merging and turning the generative adversarial network (GAN) structure with a low-dimensional semantic domain, formed the space and constructed it. It obtained 8% improvement on the Argoverse validation dataset baseline. They therefore sampled the estimated distribution from this space in a way which helped the method to monitor the representation of semantically different scenarios.

A CNN and state-transitive LSTM-based approach was demonstrated with multi-auxiliary tasks for retrieving dynamic temporal information from different driving scenarios to estimated steering angles and velocity simultaneously [158]. The method applied the vehicle's current location to determine the end-to-end driving model sub-goal angle to boost the steering angle estimation accuracy, which forecasted that the efficiency of the driving model would improve significantly. The combined method obtained 2.58° and 3.16° MAE for steering angle prediction and 0.66 m/s and 0.93 m/s speed MAE in GTA

V and Guangzhou Automotive Cooperate datasets, respectively. Nevertheless, it showed a slow response in unknown environment, so this method might not be applicable in practical implementation.

In a similar manner, Toromanoff et al. presented a CNN-based model for lateral control of AVS using a fisheye camera with label augmentation technique for accurate corrections labelling under lateral control rule to tackle ceases of lateral control error in wide FoV [159]. This method compares with pure offline methods where feedback was not implemented from a prediction which resulted in 99.5% and 98.7% autonomy in urban areas and highways after training with 10,000 km and 200 h driving video.

On the other hand, Smolyakov et al. reduced a huge number of parameters of CNN to avoid overfitting along with helping to find dependency on data sequence and implement in a CarND Udacity Simulator for predicting steering angles. However, the obtained unsatisfactory result was comparable to other reviewed results, where the accuracy was 78.5% [160].

Similarly, a CNN-based approach was applied for both lateral and longitudinal motion controlling of AVS obtaining 100% autonomy on e-road track on TORCS simulator. Although it had performed very well, contributing to both kinds of motion controlling, it lacked training data for practical implementation and memory consumption for training two different neural networks for speed and steering angle prediction. This method could be better approached by implementing in real scenarios with a good amount of training data [161].

In another proposal, a reinforcement learning-enabled throttle and brake control system was proposed by Zhu et al. [162], focusing on a one leader and one follower formation. A neural dynamic programming algorithm evaluating with trial-and-error method was directly applied for adopting near-optimal control law. The control policy included the necessary throttle and brake control commands for the follower according to the timely modified corresponding condition. Simulation experiments were carried out using the well-known CarSim vehicle dynamic simulator to show the reliability of the approach provided.

To overcome traditional sensor-based pipeline for controlling AVS where there is a tendency to learn from direct mapping, Xiao et al. demonstrated multimodal end-to-end AVS applying conditional imitation learning (CIL), taking an RGBD image as raw data in a Carla simulator environment [163]. The CNN-based CIL algorithm was evaluated in different weather modes to identify the performance for end-to-end control. The success rate of controlling in one turn and dynamic environment were 95% and 84%, respectively, which could be boosted through early fusion by changing the number of color channels from three (RGB) to four (RGBD). However, performance dropped almost 18.37% and 13.37% during controlling AVS with RGB image input for one turn and dynamic environment, respectively, in a new map of Carla simulators which could be considered as uncertain area

In brief, most of the deep learning approaches for end-to-end controlling and motion predications were based on CNN, showing efficient outcomes suitable for practical level 4 or 5 AVS. Most of the methods were deployed for estimating continuous steering angle and velocity, some controlling approaches taking into account resolving blind spot, gap estimation, overcoming slow drifting, both lateral and longitudinal motion controlling with methods such as multimodal multitask-based CNN, CNN-LSTM, Deep ConvNet, ST-LSTM, neural dynamic programming-based reinforcement learning with actor-critic network and RL. These methods faced challenges, such as noise created by human factor reasoning speed changes causing lower accuracy, only equipped by focusing on the ground truth of the current frame steering angle and not applying in a practical or complex environment. The overall summary of discussed methods is presented in Table 11.

**Table 11.** Summary of multiple deep learning methods for end-to-end controlling and prediction.

| Ref. | Method | Outcomes | Advantages | Limitations |
|---|---|---|---|---|
| [144] | Hybrid weighted KNN | Gained 85.5% accuracy in 0.223 s in best case. | Performed safe control during lane changing in uncertain environment. | Unsafe driving behavior and did not consider complex conditions. |
| [146] | CNN + LSTM + State method | Successfully learnt to navigate the track at speeds of up to 7.5 m/s. | High-speed driving control and robustness to compound errors. | Trained only for elliptical racetracks with no other vehicles. |
| [147] | CNN with comma.ai | Obtained appropriate steering angles with 2.42 mean absolute error. | Able to overcome slow drifting from human driving data. | Improper dataset for practical implementation. |
| [148] | Multimodal-based CNN | Scored 1.26° MAE for angles and 0.45° MAE for velocity. | Accurate estimation continuous steering angles and velocity. | Noise of human factor for speed changes caused lower accuracy. |
| [149] | CNN and LSTM | Almost similar steering prediction value as ground truth. | Resolved the limitation of CNN and blind-spot problem. | Lack of driving data collection from vehicle. |
| [150] | ConvNet | 90% autonomy value and 98% autonomous approximately. | Required fewer training data with no manual decomposition. | Robustness were not successful in internal processing phase. |
| [151] | Deep ConvNet | 0.033° and 0.025° steering angle MAE on GIST and Caltech baseline. | Considered lane gap and records of previous car driving. | Only tested on simple cases. |
| [152] | DDQN | 271.13% better than DQL for speed control. | Make optimal control increasing precision and policy efficiency. | Measured few errors on uneven roads. |
| [153] | ST-LSTM Network | Obtained 0.0637 RMSE value on Udacity with model weight. | Implemented in challenging lighting conditions. | Only focused on the ground truth of steering angle. |
| [154] | Spatiotemporal Conv-LSTM | Showed lowest RMSE value 0.0491 to predict steering angles. | Overcome the limitation of [153]. | Did not test in busy environment. |
| [156] | DNN | Showed on average 40% less RMSE retrieving steering angles. | Predicted steering angles in multiple conditions. | High developing cost for training individual DNN model. |
| [157] | GAN | 8% improvement for Argoverse validation dataset baseline. | Better trajectory building layers while motion prediction. | Had not tested in real time and only used simple semantics. |
| [158] | CNN and state-transitive LSTM | Predicted 2.58° and 3.16° MAE for steering angle. | Used current position and subgoal angle for steering angle prediction. | Slow prediction rate in unknown environment. |
| [159] | CNN | Achieved 99.5% and 98.7% accuracy in urban areas and highways, respectively. | Solved lateral controlling using fisheye camera. | Autonomy dropped while sharp turning. |
| [160] | CNN | Achieved 78.5% accuracy. | Reduced parameters of CNN to avoid overfitting on data sequence. | Noticeable performance drop. |

**Table 11.** *Cont.*

| Ref. | Method | Outcomes | Advantages | Limitations |
|---|---|---|---|---|
| [161] | CNN | Obtained 100% autonomy on e-road track on TORCS. | Showed both lateral and longitudinal motion control. | Lack of training data and memory consuming. |
| [162] | RL with ACN | Robust throttle and brake value of the host vehicle. | Learned controlling policy while following lead vehicle. | Environmental surroundings were not stated. |
| [163] | CIL | 84% success rate in dynamic environment. | Demonstrated successful multimodal approach in four cases. | Up to 18.37% performance drop in unknown map. |

### 3.4. Path and Motion Planning

Precipitation-based autonomous navigation including path and motion planning in an unknown or complex environment is one of the critical concerns for developing AVS. To tackle the current problem and analyze the contribution, multiple deep learning and deep reinforcement learning (DRL) combined methods for path and motion planning are reviewed in this section.

Initially, You et al. focused on the issue of path planning of autonomous vehicles in traffic in order to repeat decision making by replicating the optimum driving technique of expert drivers' actions for lane changing, lane and speed maintenance, acceleration and braking in MDPs on highways [164]. The optimal control policy for the proposed MDP was resolved using deep inverse reinforcement learning (DIRL) and three MaxEnt IRL algorithms by utilizing a reward function in terms of a linear combination of parameterized function to solve model-free MDP. The trajectory proposals were executed at the time of overtaking and the policy recovery was reduced to 99%, even though there was insufficient evidence for the reflection of stochastic behavior.

To solve limitations of rule-based methods for safe navigation and better intersection problems for AVS, a vision-based path and motion planning formula was used by Isele et al., adopting DRL [165]. Each wait action was proceeded by another wait or go action, meaning that each pathway was a series of waiting decisions that concluded in a go decision as well as the agent not being permitted to wait after the go action had been chosen. The method secured a success rate for forward, right, left and turn and challenge of 99.96%, 99.99%, 99.78% and 98.46%, respectively, which was 28% faster than the TTC (time-to-collision) method, although performance decreased three times and average time doubled during this challenging situation.

Zhang et al. proposed a risk analysis and motion planning system for autonomously operated vehicles focused on highway scenario motion prediction of surrounding vehicles [166]. An interactive multiple model (IMM) and constant turn rate and acceleration (CTRA) model were used for surrounding vehicle motion prediction, and model predictive control (MPC) was used for trajectory planning that scored 3.128 RMSE after 5 s during motion prediction. Although it was designed for connected AVS, it is efficient for vision-based approaches.

Another approach, local and global path planning methodology, was presented in an RoS-based environment for AVS by Marin-Plaza et al., where they used the Dijkstra and time elastic bands (TEB) method [167]. The path planning model was able to reach the goal with modest error by calculating Euclidean distance for comparing local and global pan waypoints, where it scored 1.41 m, which is very efficient. However, it was applicable only if the model was not specifically calibrated for the vehicle's kinematics or if the vehicle was out of track, and did not consider complex scenarios. In another work, Islam et al. established a vision-based autonomous driving system that relied on DNN, which handled a region with unforeseen roadway hazards and could safely maneuver the AVS in this environment [168]. In order to overcome an unsafe navigational problem, they presented object detection and structural segmentation-based deep learning architecture, where it

obtained an RMSE value of 0.52, 0.07 and 0.23 for cases 1 to 3, respectively, and 21% safety enhancement adding hazard avoiding method.

Ma et al. proposed an efficient RRT algorithm that implemented a policy framework based on the traffic scenes and an intense search tree extension strategy to tackle traditional RRT problems where it faced a meandering route, an unreliable terminal state and sluggish exploration, and established more sustainable motion planning for AVS [169]. In addition, the integrated method of the proposed fast RRT algorithm and the configuration time space could be adopted in complex obstacle-laden environments to enhance the efficiency of the expected trajectory and re-planning. A significant set of experimental results showed that the system was much quicker and more successful in addressing on-road autonomous driving planning queries and demonstrating its better performance over previous approaches.

In another work, an optimum route planner integrated with vehicle dynamics was designed by Gu et al. implementing an artificial potential field to provide maximum workable movement that ensured the stability of the vehicle's path [170]. The obstacles and road edges were typically used with constraints and not with any arbitrary feature in this method in the optimal control problem. Therefore, when designing the optimum route using vehicle dynamics, the path-planning method was able to treat various obstacles and road structures sharply in a CarSim simulator. The analysis showed that the method reduced computational costs by estimating convex function while path planning. A similar method was proposed by Wahid et al., where they used an artificial potential field with adaptive multispeed scheduler for a collision-avoidance motion planning strategy [171].

Cai et al. demonstrated a novel method combining CNN, LSTM and state model which was an uncertainty-aware vision-based trajectory generation network for AVS's path-planning approach in an urban traffic scene [172]. The work was divided into two major parts: the first one was a CNN bottleneck extractor, and the second component included a self-attention module for calculating recurrent history and an LSTM module for processing spatiotemporal characteristics. Finally, they designed the probable collision-free path planning with speeds and lateral or longitudinal locations for the next 3.0 s after taking image stream and state information in the past 1.5 s considering as input. The method obtained more centralized error distribution and lower error medium.

For safe navigation for AVS in road scenarios with obstacles, a model prediction control-based advanced dynamic window (ADW) method was introduced by Kiss et al. [173]. The method demonstrated differential drive that reached the destination location ignoring the desired orientation and did not require any weighted objective function.

A motion planning model based on the spatiotemporal LSTM network (SLN), which had three major structural components, was proposed by Bai et al. It was able to produce real-time feedback based on the extraction of spatial knowledge [174]. First, convolutional long-term memory (Conv-LSTM) was applied in sequential image databases to retrieve hidden attributes. Secondly, to extract spatiotemporal information, a 3D CNN was used, and precise visual motion planning was displayed constructing a control model for the AV steering angle with fully connected neural networks. The outcome showed almost 98.5% accuracy and better stable performance compared with Hotz's method [147]. Nonetheless, the method was found to minimize state after generating overfitting on antecedent data for time-series data of previous steps, causing more computational cost and time.

Another motion planning avoiding-obstacle-based approach was proposed in a simulation environment [175]. The motion planning method had the ability to infer and replicate human-like control thinking in ambiguous circumstances, although it was difficult to establish a rule base to tackle unstructured conditions. The approach was able to execute 45.6 m path planning with 50.2 s.

In conclusion, very few works have adopted a perception-based path and motion planning for AVS but the existing research adopting deep inverse reinforcement learning and MaxEnt IRL, deep Q-network time-to-go method, Dijkstra and time elastic bands method, DNN, advance RRT, artificial potential field, ADW using model predictive control and fuzzy logic made a remarkable contribution, with high accuracy, collision-free path

planning, 21% safety enhancement adding hazard-avoiding method planning motion in a multilane turn-based intersection. Nevertheless, these methods were not practically implemented or theoretical, and some of the high-performing approaches were not tested in a real-life environment with heavy traffic. An overview of the deep learning methods selected for analysis to improve AVS is presented in Table 12.

**Table 12.** Summary of multiple deep learning methods for path and motion planning.

| Ref. | Method | Outcomes | Advantages | Limitations |
|------|--------|----------|------------|-------------|
| [164] | DIRL | 99% policy recovery within less data length. | Avoid cost function and manual labelling. | Insufficient training data for stochastic behavior representation. |
| [165] | DRL with TTC | 28% faster than TTC method. | Solved limitation of rule-based intersection problems. | Performance decreased three times during challenging situation. |
| [166] | IMM with MPC | Score 3.128 RMSE after 5 s during motion prediction. | Considered motions of surrounding vehicles. | Lower accuracy in far predicted horizon. |
| [167] | Dijkstra and TEB method | Obtained efficient Euclidean distance 1.41 m. | Reach goal with modest error. | Applicable only if vehicle was out of track in simple scenarios. |
| [168] | DNN | 21% safety enhancement adding hazard-avoiding method. | Safe navigation adding hazard detection and segmentation method. | Tested only on simple open highway. |
| [169] | Advance RRT | Took 5 ms and 48 ms for path selection and path generation. | Novel cost function to select path and obstacle-avoiding feature. | Limited to non-rule-based approach. |
| [170,171] | Artificial potential field | Visualized potential field in nine different scenarios. | Reduce computational cost by estimating convex function. | Effects of local minimum issue that led AV to be stuck in a position. |
| [172] | CNN + LSTM + State method | Lower error medium and more centralized error distribution. | Vehicle motion planning predicted in multilane turn-based intersection. | Did not consider traffic light and weather condition for performance evaluation. |
| [173] | ADW with MPC | Reached destination location ignoring the desired orientation. | Did not require any weighted objective function. | Limitation occurred with constrained kinematics. |
| [174] | 3D-CNN | Almost 98.5% average accuracy and stable outcome. | Able to learn time-serial features from traffic environment. | Minimized state after generating overfitting on time-series data. |
| [175] | Fuzzy logic | 45.6 m path planning with 50.2 s. | Human-like control thinking in ambiguous circumstances. | Difficult to establish a rule base to tackle unstructured conditions. |

### 3.5. AR-HUD

Augmented reality (AR) in head-up display (HUD) or displaying in windshield for autonomous driving system as a medium of final visualizing of activities outcomes from the deep learning approach was overlayed with an autonomous driving system. The AR-based vehicular display system was essential for driving situation awareness, navigation and overall deployment as a user interface.

Yoon et al. demonstrated an improved forward collision alert system detection of cars and pedestrians fused into the HUD with augmented reality through using stereo cameras and visualized early alerts where SVM classifier was applied for object recognition and obtained an F1 score of 86.75% for car identification and 84.17% for pedestrian iden-

tification [176]. The limitation of the work was noticed when the observed object moved rapidly and the car suddenly turned; it was visualized with delay. The proposed system yet needed to optimize efficiency and acceleration which in diverse vehicle conditions responds robustly to different and high speeds.

An analysis showed personal navigation with AR navigation assist equipped for use with a volumetric 3D-HUD and utilizing its parameters. An interface was developed for assisting to turn faster by locating turn points quicker than during regular navigation [177]. The interface also helped to maintain user eyes and to fix them more precisely on the driving environment after analyzing traffic scenes with deep learning algorithm with proper registration of applications via spatial orientation of AR views on interface. On the basis of the results, however, the inadequate perception of the depth of a specified 2D HUD distance is obvious and the navigation system's AR interface was ineffective without a 3D HUD.

An automatic AR based on a road tracking information method registration was introduced by Yoon et al., with a SIFT matching function and homography measurement method, which defined matching between camera and HUD providing the driver's view was positioned to the front, which detected vehicle and pedestrians and converted them into AR contents after projective transformation [178]. This solution was good enough for daytime performance but had limitations at nighttime. Nevertheless, the procedure had the ability to automate the matching without user interference, but it is inconvenient while projecting outcomes which occurred due to misreading local correspondence.

Park et al. demonstrated an AR-HUD-based driving safety instruction by identifying vehicle and pedestrians using the INRIA dataset [179]. The identification method was built using SVM and HOG with 72% and 74% in fps accuracy and detected partial obstacles, respectively, applying a billboard sweep stereo (BSS) algorithm. The detected vehicles and pedestrians were overlapped on the HUD with the AR technique. Despite detecting obstacles in sunny and rainy scenarios, it was not deployed for nighttime scenarios.

In order to integrate outcomes with AR, the system was divided into two parts by Rao et al., 3D object detection and 3D surface reconstruction, to develop object-level 3D reconstruction using Gaussian Process Latent Variable Model (GPLVM) with SegNet and VPNet for in-vehicle augmented reality UI and parking system [180]. Their AR-based visualization system was built with monocular 3D shaping, which was a very cost-efficient model and needed only a single frame in the input layer.

Furthermore, a new traffic sign-recognition framework based on AR was constructed by Abdi and Meddeb to overlay traffic signs with more recognizable icons overlapped in an AR-HUD to increase the visualization of a driver aiming to improve safety [181]. The Haar Cascade detector and the verification of the theory using BoVW were combined with the relative spatial data between visual words, which had proven to be a reasonable balance between resource efficiency and overall results. A classifier with an ROI and allocated 3D traffic sign was subsequently developed using a linear support vector machine that required less training and computation time. During the decision-making process, this state-of-the-art methodology influenced the distribution of visual attention and could be more consistent with the improved approach of deep learning recognition relying on the GPU.

Addressing the challenge of overtaking an on-road slow vehicle, a see-through effect-based marker-less real-time driving system had been demonstrated by Rameau et al., applying AR [182]. To overcome the occlusion and produce a seamless see-through effect, a 3D map of the surroundings was created using an upper-mounted camera and implementing an in-vehicle pose predictor system. With up to 15 FPS, they presented a faster novel real-time 2D–3D tracking strategy for localization of rear in a 3D map. For the purpose of decreasing bandwidth usage, the ROI was switched to the rear car impacted by an occlusion conflict. This tracking method on AR-HUD showed great efficiency and easy adoption capability for vehicle displaying systems.

To reduce the accident cases, Abdi et al. proposed augmented reality-based head-up display providing more essential surrounding traffic data as well as increasing interactions between drivers and vehicles to enhance drivers' focus on the road [183]. A custom deep CNN architecture was implemented to identify obstacles and final outputs will be projected in the AR head-up display. For AR-based projection in HUD, firstly, pose prediction of targeted ROIs were carried out and obtained 3D coordinates with points after achieving camera projection matrix to recognize AR 3D registration. This step created a 6-DOF pose of translation and rotation parameters which will be helpful for motion estimation calculation with planar homograph. Afterwards, the RANSAC method was applied to compute the homograph matrix, and OpenGL real camera was synchronized with a virtual camera that showed a projection matrix to map 2D points utilizing 3D surface points and developed a marker-less approach.

Lindemann et al. demonstrated an augmented reality-based windshield display system for autonomous vehicle with a view to assisting driving situation awareness in city areas and increase automated driving level from level 4 to 5 [184]. This AR-based windshield display UI was developed based on deep learning-applied object detection to enhance situation awareness, aiming at both clear and lower-visibility conditions where they obtained very different situation awareness scores in low-visibility conditions in disabled windshield display but failed to obtain a good score when windshield UI was enabled. Nevertheless, it worked significantly better in clear weather conditions.

Park et al. presented a 2D histogram of oriented gradient (HOG) tracker and an online support vector machine (SVM) re-detector based on training of the TLD (tracking-learning-detector) functional vehicle tracking system for AR-HUD using equi-height mosaicking image (EHMI) [185]. The system initially performed tracking on the pre-computed 2D HOG EHMI, when the vehicle was identified in the last frame. If the tracking failed, the system started re-detection using an online learning-based SVM classification. The tracking system conducted online learning frequently after the vehicle had been registered and minimized the further calculation necessary for tracking as the HOG descriptor for EHMI was already determined in the detection phase. The technique was perfect for deploying in various lighting and occlusion scenes since it adopted online learning. Refining the algorithm to make optimized hardware or embedded device and to identify other dangerous obstacles effectively in road scenes, this lightweight architecture-based proposed work could be a more acceptable approach for faster tracking and visualizing in HUD.

To represent driving situation awareness data, Park et al. introduced a vehicle augmented-reality system that deducts drivers' distractions with an AR-based windshield of the Genesis DH model from Hyundai motors [186]. The system presented driving conditions and warned a driver using a head-up monitor via the augmented reality. The system included a range of sub-modules, including vehicle and pedestrian recognition based on the deep learning model of [179], vehicle state data, driving data, time to collision (TTC), hazard evaluation, alert policy and display modules. During most experiments, on the basis of TTC values and driver priority, the threat levels and application of augmented EHMI was already determined in the detection phase.

In this section, a combination of deep learning algorithms and their outcomes were visualized as the final task of AVS showing them in an AR-based HUD for better driving assistance. AR-HUD was adopted due to visualization in front display for early warning, navigation, object marking by overlapping, ensuring safety and better tracking. Although these studies had successful demonstrations, some major limitations were detected when analyzing the studies, such as visual delay for the case of sudden turn or rapid-moving objects, misreading of local correspondence, high computational cost while 3D shaping, visualizing challenges in extreme contrast and distraction for complex UI. Table 13 provides a summary of the section.

**Table 13.** Summary of multiple deep learning methods for AR-HUD.

| Ref. | Purpose | Methods | Advantages | Limitations |
|------|---------|---------|------------|-------------|
| [176] | Early warning | SVM | Improved collision alert system detecting cars and pedestrians fused into the HUD. | Visualization delay while observing rapid moving and sudden turning of vehicle. |
| [177] | Navigation | Custom deep learning based scene analysis. | Helped to turn faster and more confidently locating turn points quicker. | The insufficient depth perception of the defined 2D HUD distance was apparent. |
| [178] | Object marking | SIFT and homography measurement method. | Detected road objects are converted into AR contents after projective transformation. | Automatic matching ability is inconvenient due to misreading of local correspondence. |
| [179] | Safety | SVM, HoG and BSS algorithm. | Applicable in sunny and rainy scenarios for overlapping of detected objects and obstacles. | Poor detection accuracy and not applicable for nighttime scenarios. |
| [180] | Assistance | GPLVM with SegNet and VPNet. | 3D shaping cost-efficient model and needs only single frame in input layer. | Computational complexity is higher for algorithm fusion. |
| [183] | Safety | Haar cascade detector with BoVW method. | Overlay traffic signs with more recognizable icons in AR-HUD to improve safety. | Lack of implementation in complex scenarios. |
| [182] | Tracking | 2D–3D tracking strategy with ROI. | Assist overtaking an on-road slow vehicle via marker-less real-time driving system. | The detection methods for deployment in the real-life field had yet to become apparent. |
| [183] | Safety | CNN and RANSAC. | Providing more essential surrounding traffic data to increase interactions and focus. | Was not deployed in complex traffic scenarios or nighttime environments. |
| [184] | Safety | Custom deep learning applied object detection. | Boost situation awareness aiming at both clear and lower-visibility conditions | Failed to achieve good visualization in lower-visibility conditions. |
| [185] | Tracking | TLD using SVM. | Applicable in various lighting and occlusion scenes, since it adopted online learning. | Required a very wide range of views. |
| [186] | Safety and awareness | SVM and HoG. | Enhanced a driver's intuitive reasoning and minimized driver distraction calculating TTC. | Not computation cost efficient and complex UI. |

## 4. Evaluation Methods

In this section, commonly used evaluation metrics throughout the systematic review are presented in Table 14. Several evaluation techniques with equations and description are shown which will give a better understanding, as evaluation techniques are different from the reviewed methodology.

**Table 14.** Evaluation techniques.

| Ref. | Technique | Equations | Remarks |
|------|-----------|-----------|---------|
| [40,44,50,53,72,76,82,90, 110,116,119,120,137] | Sensitivity/TPR/Recall (*R*) | $R = \frac{TP}{TP+FN}$ | *TP* is the true positive and *FN* is the false negative detection. |
| [40,44,67,72,90,110,116, 119,120] | Precision (*P*) | $P = \frac{TP}{TP+FP}$ | *FP* is the false positive detection. |
| [54,80,88,90,110,119,120, 176] | F1 Score | $F1 = 2 * \left( \frac{R \times P}{R+P} \right)$ | - |
| [37–39,41,43,46– 49,52,57,60,61,64,79,91– 94,96,98–101,106,108,112– 114,118,120,122,137,142, 144,152,179] | Accuracy | $Accuracy = \frac{X_{Pred}}{X_{GT}}$ | $X_{Pred}$ is the number of successes and $X_{GT}$ is the ground truth. |
| [95,103–105,111,112] | mIoU | $IoU = \frac{Region\ of\ intersection}{Region\ of\ union}$ <br> $mIoU = \frac{1}{k+1} \sum\limits_{n}^{k} \frac{TP}{TP+FP+FN}$ | - |
| [34,63,77] | FNR | $FNR = \frac{FN}{TP+FN}$ | - |
| [35,42,45,56,73,75,92] | mAP | $mAP =$ <br> $\frac{1}{n} \sum\limits_{k=1}^{k=n} \left( (R(k) - R(k+1)) * P \right)$ | *k* denotes each episode and *n* is the total episodes. |
| [65,66,68,76,78] | Log Average Miss Rate ($Log_{AMR}$) | $Log_{AMR} = exp\left( \frac{1}{n} \sum\limits_{i=1}^{n} \ln a_i \right)$ | $a_i$ is the series positive values correlated with the missing rate. |
| [51,55,81,116] | Area Under Curve (*AUC*) | $AUC = \int TPR\ d(FPR)$ | *TPR* is the true positive rate, and *FPR* is the false positive rate. |
| [83,89,131] | Lateral Error | $\Delta y = \Delta y_r + (L * \varepsilon_r)$ | $\Delta y_r$ is center of gravity, $\varepsilon_r$ is yaw angle towards the road and *L* is the distance. |
| [121,138,139,142] | Reward (*r*) | $r = \begin{cases} min(t_1, t_2) \\ if\ l = 1\ and\ d_1, d_2 > 3 \\ min(t_3, t_4) \\ if\ l = 2\ and\ d_1, d_2 > 3 \\ -5\ else \end{cases}$ <br> $where,\ t_n = \frac{d_n - d_{sn}}{v_n}$ | $t_n$ is dimension of state, $d_n$ is measured distance, $d_{sn}$ is safe distance, *l* is overtaking lane and $v_n$ is the longitudinal velocity. Here, *n* = 1,2 refers host and lead vehicle in driving lane and *n* = 3,4 refers to overtaking lane. |
| [123,126,128,130,136] | Collision Rate ($C_{rate}$) | $C_{rate} = \frac{N_{col}}{C_{lap}}$ | $N_{col}$ is the total collision number while completing total $C_{lap}$ laps. |
| [102,124,127,134,165] | Success Rate (*SR*) | $SR = \frac{Success\ Counts}{Total\ Test\ Number} * 100\%$ | - |
| [129,135] | Safety Index (*SI*) | $\bar{s} = \frac{N_{wc}}{N_{ep}}$ | $N_{wc}$ is total episode without collision and $N_{ep}$ is the total episodes. |
| [133,147,148,151,158] | MAD/MAE | $MAE/MAD = \frac{1}{n} \sum\limits_{i=1}^{n} |x_i - \hat{x}_i|$ | $x_i$ and $\hat{x}_i$ are the real and predicted value, respectively, and *n* is the total episodes. |
| [137] | Specificity (*SPC*) | $SPC = \frac{TN}{FP+TN}$ | *TN* is the true negative value. |
| [150,159] | Autonomy value (*Av*) | $Av = \left( 1 - \left( \frac{N_i}{T_e} \right) * 6 \right) * 100\%$ | $N_i$ is total interventions and $T_e$ is elapsed time. |
| [153–156,166–168] | RMSE | $RMSE = \sqrt{\frac{\sum_{i=1}^{n} (x_i - \hat{x}_i)^2}{n}}$ | $x_i$ is the ground truth and $\hat{x}_i$ is the predicted truth. |
| [117] | *p*-Value | $Z = \frac{y' - y^0}{\sqrt{\frac{y^0(1-y^0)}{c}}}$ | $y'$ is the sample proportion, $y^0$ is the assumed proportion, *n* is sample size. |

## 5. Case Study

### 5.1. AVS in Roundabout Cases

A roundabout is one of the most difficult scenarios for driving with AVS due to tight time constraints on vehicle behavior because of the yield and merging of maneuvers with high-quality vision requirements for estimating the state of other vehicles, and multi-factor decision-making based on these state estimations. It is particularly tough to forecast the actions of other cars in a roundabout due to the availability of numerous exit locations as discrete options mixed with continuous vehicle dynamics. The entry risk at roundabouts grows with decreasing distance as the ego vehicle must account for cars in the circle passing. Okumura et al. proposed a neural network to map observations to actions in a roundabout that are handled as a combination of turns in order to emphasize the deep learning-based AVS for roundabout cases [197]. This method concentrated on route planning and speed estimation for the roundabout, as well as detection, tracking and generating predictions about the environment using sensor fusion, but ignored interactions between cars.

This method concentrated on route planning and speed estimation for the roundabout, as well as detection, tracking and generating predictions about the surroundings using sensor fusion, but ignored interactions between cars [198]. This could be improved by a strategy for forecasting whether a vehicle will exit the roundabout based on its anticipated yaw rate. In a roundabout scenario, the projected yaw rate is a significant indication of whether a car will turn next to the ego vehicle. Although the system was proved to be capable of human-like judgments for a certain roundabout situation, only the center of mass and velocity were calculated to quantify detection of turning cars. This method may be a viable solution for the roundabout research of [197]; however, it may result in errors in roundabouts with no traffic lights or heavy traffic.

One of the main reasons of vision-based AVS is to reduce the dependency in terms of safety and collision-free driving; therefore, combined multi-thread architecture of algorithms such as Spatial CNN (SCNN) and Deep Recurrent Q-Network (DRQN) could be a major solution for roundabout cases. The spatial features of SCNN for traffic scene understanding in dense traffic conditions as well as the ability of extreme efficient traffic scene-analysis demonstration incorporating multi-threading with self-decision making improved DRL approaches such as DRQN or DDQL could be a vast improvement in the research of AVS in roundabout cases.

### 5.2. AVS in Uncertain Environmental Cases

Even at the current development level, it is challenging for AVS to operate autonomously in unknown or uncertain environments. The uncertainty may be because of variable traffic conditions, unknown terrain, unmarked or untrained settings, or even a situation including an extended obstruction. In an unexpected driving environments, even the performance of Waymo, the self-driving vehicle of Google, is at a conditional level 4 of autonomy based on NHTSA autonomous functions, and Tesla's self-driving vehicles are only at level 2 of autonomy. In contrast, the authors of one study addressed safety issues posed by ambiguity in DL approaches: insufficient training data, locational shift, inconsistencies between training and operating parameters, and uncertainty in prediction [199]. The most controversial incident occurred when a Tesla Model S was involved in a deadly collision in which the driver was killed when its autopilot system failed to notice a tractor-trailer 18-wheeler that turned in front of the vehicle [200]. To reduce unintended occurrence in unknown or uncertain situations and environments, it might be possible to develop level 4 or 5 AVS with safe perception analysis, path planning, decision making and controlling by removing dependence on labelled data and adopting deep reinforcement learning-based approaches. Moreover, several techniques, such as those in [83,128,130,144,172], which were effective in avoiding collision, lane shifting, detection, and safe decision making in unknown or dynamic situations, can be a means of reducing the constraints in uncertain environments.

## 6. Discussion

Deep learning is fast becoming a successful alternative approach for perception-based AVS as it reduces both cost and dependency on sensor fusion. With this aim in mind, total categories of primary domains of AVS were reviewed in this paper to identify efficient methods and algorithms, their contributions and limitations.

From the study, it was found that recent deep learning algorithms obtained high accuracy while detecting and identifying road vehicle types, and in some cases, the results surpassed LiDAR's outcome in both short and long range for 3D bounding vehicles [34]. Moreover, some recent methods such as YOLO V2 [35], deep CNN [38], SINET [41] and Faster R-CNN [42] achieved high accuracy within a very short period of time from low-quality training images to challenging nighttime scenarios. However, there were several limitations, for example, in certain lighting conditions and higher execution costs. Following that, a massive contribution to lane and curve detection along with tracking was presented by studies where 95.5% road scene extraction was demonstrated, for example, in [79], for lane edge segmentation without manual labelling using a modified CNN architecture. As discussed in previous sections, challenges such as higher computational cost [81], insufficient for far field of view [82], not testing in complex scenarios [79] and poor luminance made some proposals tough for practical implementation in present AVS.

In addition, a good amount of attention was given to developing safe AVS systems for pedestrian detection. Multiple deep learning approaches such as DNN, CNN, YOLO V3-Tiny, DeepSort R-CNN, single-shot late-fusion CNN, Faster R-CNN, R-CNN combined ACF model, dark channel prior-based SVM, attention-guided encoder–decoder CNN outperformed the baseline of applied datasets that presented a faster warning area by bounding each pedestrian in real time [61], detection in crowded environments, and dim lighting or haze scenarios [62,72] for position estimation [72], minimizing computational cost and outperforming state-of-the-art methods [120]. The approaches offer an ideal pedestrian method once their technical challenges have been overcome, for example, dependency on preliminary boxing during detection, presumption of constant depths in input image and improvement to avoid missing rate when dealing with a complex environment.

Moreover, to estimate steering angles, velocity alongside controlling for lane keeping or changing, overcome slow drifting, take action on a human's weak zone such as a blind spot and decreasing manual labelling for data training, multiple methods, such as multimodal multitask-based CNN [148], CNN with LSTM [149] and ST-LSTM [153], were studied in this literature review for AVS's end-to-end control system.

Furthermore, one of the most predominant segments of AVS, traffic scene analysis, was covered to understand scenes from a challenging and crowded movable environment [102], improve performance by making more expensive spatial-feature risk prediction [112] and on-road damage detection [120]. For this purpose, HRNet + contrastive loss [104], Multi-Stage Deep CNN [106], 2D-LSTM with RNN [108], DNN with Hadamard layer [110], Spatial CNN [112], OP-DNN [113] and the methods mentioned in Table 9 were reviewed. However, there are still some limitations, for instance, data dependency or relying on pre-labelled data, decreased accuracy in challenging traffic or at nighttime.

Taking into account all taxonomies as features, the decision-making process for AVS was broadly analyzed where driving decisions such as overtaking, emergency braking, lane shifting with collision and driving safety in intersections adopting methods such as deep recurrent reinforcement learning [127], actor-critic-based DRL with DDPG [123], double DQN, TD3, SAC [124], dueling DQN [126], gradient boosting decision tree [133], deep RL using Q-masking and autonomically generated curriculum-based DRL [139]. Despite solving most of the tasks for safe deployment in level 4 or 5 AVS, challenges remain, such as complex training cost, lack of proper surrounding vehicles' behavior analysis and unfinished case in complex scenarios. Some problems remain to be resolved for better outcomes, such as the requirement of a larger labelled dataset [57], struggle to classify in blurry visual conditions [49] and small traffic signs from a far field of view [51], background complexity [48] and detecting two traffic signs rather than one, which occurred for different

locations of the proposed region [47]. Apart from these, one of the most complicated tasks for AVS, only vision-based path and motion planning were analyzed by reviewing approaches such as deep inverse reinforcement learning, DQN time-to-go method, MPC, Dijkstra with TEB method, DNN, discrete optimizer-based approach, artificial potential field, MPC with LSTM-RNN, advance dynamic window using, 3D-CNN, spatio-temporal LSTM and fuzzy logic, where solutions were provided by avoiding cost function and manual labelling, reducing the limitation of rule-based methods for safe navigation [164] and better path planning for intersections [165], motion planning by analyzing risks and predicting motions of surrounding vehicles [166], hazard detection-based safe navigation [168], avoiding obstacles for smooth planning in multilane scenarios [169], decreasing computational cost [170] and path planning by replicating human-like control thinking in ambiguous circumstances. Nevertheless, these approaches faced challenges such as lack of live testing, low accuracy in far predicted horizon, impaired performance in complex situations or being limited to non-rule-based approaches and constrained kinematics or even difficulty in establishing a rule base to tackle unstructured conditions.

Finally, to visualize overlaying outcomes generated from the previous methods superimposed on the front head-up display or smart windshield, augmented reality-based approaches combining deep learning methods were reviewed in the last section. AR-HUD based solutions such as 3D surface reconstruction, object marking, path overlaying, reducing drivers' attention, boosting visualization in tough hazy or low-light conditions by overlapping lanes, traffic signs as well as on-road objects to reduce accidents using deep CNN, RANSAC, TTC methods and so on. However, there are still many challenges for practical execution, such as human adoption of AR-based HUD UI, limited visualization in bright daytime conditions, overlapping non-superior objects as well as visualization delay for fast moving on-road objects. In summary, the literature review established for vision-based deep learning approaches of 10 taxonomies for AVS with discussion of outcomes, challenges and limitations could be a pathway to improve and rapidly develop cost-efficient level 4 or 5 AVS without depending on expensive and complex sensor fusion.

## 7. Conclusions

The results of the mixed method studies in the field of implementation and application of deep learning algorithms for autonomous driving systems help us to achieve a clear understanding of the future of transportation. These results prove that it has the ability to provide intelligent mobility for our constantly evolving modern world as deep learning was one of the key components to resolve the limitations and bottlenecking of traditional techniques. Despite containing a good number of studies on autonomous driving systems, only a few make an impact on recent developments in the autonomous driving industry. To overcome this challenge and build a safer and more secure sensor-independent transportation system with the aim of building infrastructure of futuristic smart cities, in this paper, through a systematic review of the literature, studies of AV were selected that used deep learning and the field was reviewed in terms of decision making, path planning and navigation, controlling, prediction and visualizing the outcomes in augmented reality-based head-up displays. We analyzed the existing proposal of deep learning models in real-world implementation for AVS, described the methodologies, designed the flow of solutions for the limitations of other methodologies, and compared outcomes and evaluation techniques. Nevertheless, as the research field of autonomous driving systems is still growing, many of the theoretical methodologies were not applied practically, but along with the research trend of this expanding field, these are potentially excellent solutions that require further development. Thus, the large-scale distributions of the paper in the major areas of autonomous driving systems will be essential for further research and development of the autonomous vehicle industry into a cost-efficient, secure intelligent transport system.

**Funding:** This research was funded by Universiti Kebangsaan Malaysia [GUP-2020-060].

**Institutional Review Board Statement:** Not applicable.

**Informed Consent Statement:** Not applicable.

**Data Availability Statement:** Not applicable.

**Conflicts of Interest:** The authors declare no conflict of interest.

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
