# Peer review of "Vision-Based Autonomous Vehicle Systems Based on Deep Learning: A Systematic Literature Review"

_applsci, doi:10.3390/app12146831_

Round 1

Reviewer 1 Report

Comments

The authors of the manuscript have presented literature review(2011-2021) of deep learning algorithms for vision based autonomous systems specific to RGB camera vision. Since it is a broad area of research the authors have divided the research according to different modules as per research practice. While the literature review is comprehensive, the following observations/suggestions need to be incorporated.

1.      Please get the paper reviewed by a native English speaker first. The manuscript contains a lot of grammatical/structural mistakes starting with the first line of the abstract, “Autonomous vehicle system (AVS) is a trendy science and technology research areas,”. Apart from many more, see line 36 introduction section the word time consuming doesn’t make sense. Please consider revising the language used in paper as a major point for revision.

2.      Please enhance image quality for instance text in Figure 01 is barely readable.

3.      While going through google scholar search several highly cited or relevant publications were found which are not considered in the current study. This was done using basic search. The study missed several key contributions made in the field. Such quality papers need to be considered in the current survey. The paper search criteria needs to be revised.

Zablocki, É., Ben-Younes, H., Pérez, P. and Cord, M., 2021. Explainability of vision-based autonomous driving systems: Review and challenges. arXiv preprint arXiv:2101.05307.

Boukerche, A. and Ma, X., 2021. Vision-based Autonomous Vehicle Recognition: A New Challenge for Deep Learning-based Systems. ACM Computing Surveys (CSUR)54(4), pp.1-37.

Muhammad, K., Ullah, A., Lloret, J., Del Ser, J. and de Albuquerque, V.H.C., 2020. Deep learning for safe autonomous driving: Current challenges and future directions. IEEE Transactions on Intelligent Transportation Systems22(7), pp.4316-4336.

Kuutti, S., Bowden, R., Jin, Y., Barber, P. and Fallah, S., 2020. A survey of deep learning applications to autonomous vehicle control. IEEE Transactions on Intelligent Transportation Systems22(2), pp.712-733.

Mozaffari, S., Al-Jarrah, O.Y., Dianati, M., Jennings, P. and Mouzakitis, A., 2020. Deep learning-based vehicle behavior prediction for autonomous driving applications: A review. IEEE Transactions on Intelligent Transportation Systems23(1), pp.33-47.

Nguyen, H., Kieu, L.M., Wen, T. and Cai, C., 2018. Deep learning methods in transportation domain: a review. IET Intelligent Transport Systems12(9), pp.998-1004.

4.      Lastly, consider including a case study or at least, papers should be ranked according to certain relevant criteria/parameters. Data sets used in the papers should also be taken into account.

Author Response

Response to Reviewer – 01

  1. Please get the paper reviewed by a native English speaker first. The manuscript contains a lot of grammatical/structural mistakes starting with the first line of the abstract, “Autonomous vehicle system (AVS) is a trendy science and technology research areas,”. Apart from many more, see line 36 introduction section the word time consuming doesn’t make sense. Please consider revising the language used in paper as a major point for revision.

Response 1. Thank you for your kind suggestions to improve the review paper. To tackle grammatical and structural mistakes, the article is revised thoroughly and restructured many sentences to enhance the writing quality. I made sure that there would be a significant change in the revised paper with no grammatical mistakes.

  1. Please enhance image quality for instance text in Figure 01 is barely readable.

Response 2. Thank you for pointing out the lacking. Figure 01 is re-drawn with 300 PPI (High) resolution, and the same goes for the other images. The readability and visibility were checked before submitting the revised paper.

  1. While going through google scholar search several highly cited or relevant publications were found which are not considered in the current study. This was done using basic search. The study missed several key contributions made in the field. Such quality papers need to be considered in the current survey. The paper search criteria needs to be revised.

Response 3. I appreciate your kind suggestion. We tried our label best to find the most suitable and highly cited papers although unfortunately, some of the papers are missing. However, several major impactful papers with high citation were reviewed in the paper in each domain and some of the recent but highly cited papers are reviewed (for instance, [21], [34], [110], [111], [147], and [150] were added to review and discussed in Table 2). The search and selection criteria are revised as suggested (2.4 & 2.5). Restructuring  Figure 1 (Framework for literature searching and selection) was also a part of this correction.

  1. Lastly, consider including a case study or at least, papers should be ranked according to certain relevant criteria/parameters. Data sets used in the papers should also be taken into account.

Response 4. As the reviewer suggested, new sub-sections (5.1 & 5.2) are included where case studies were discussed and analysed (AVS in roundabout and uncertain cases).

Furthermore, Figure 5 is visualized to show the major applied datasets for domain/subdomains. Details of the dataset are discussed in each section while reviewing the outcomes of the adopted datasets.

Reviewer 2 Report

  • There is no mention of Table 2 in the main text.
  • Figure 2.7 is repeated (188 and 197). Must be Figure 2.8.
  • It is recommended to improve the quality of figure 1. Also make the font a little more readable.
  • I recommend, Table 4 should optimize the style regarding the sources:
  • 34, 36-46;
  • 48-50, 53-60;
  • 64,65, 67-72 and etc. all (The same is with Table 14).

Author Response

Response to Reviewer – 02

  1. There is no mention of Table 2 in the main text. Figure 2.7 is repeated (188 and 197). Must be Figure 2.8

Response 1. Thank you for the positive response to the article.  The missing mentioning Table 2 is added to the introduction session.  Moreover, I appreciate for pointing out the repentance of the two sections. It had been corrected.

  1. It is recommended to improve the quality of figure 1. Also make the font a little more readable.

Response 2. Thank you for pointing out the lacking. Figure 01 is re-drawn with 300 PPI (High) resolution.

  1. I recommend, Table 4 should optimize the style regarding the sources:

34, 36-46;

48-50,

53-60;

64,65,

67-72 and etc. all (The same is with Table 14).

Response 3. I appreciate your valuable suggestion. Table 4 is optimized as advised. However, on the contrary, this is hardly in sequence in Table 14.

Reviewer 3 Report

The paper conducts a systematic literature review of vision-based autonomous vehicle systems. It summarizes algorithms and evaluation protocols, and provides some discussions about future developments. However, some issues should be addressed before acceptance:

A detailed taxonomy of the algorithms in terms of tree or figure should be provided. That will help understanding how different algorithms or tasks can be tied together.

Some recent advances in the field should be also discussed. For example, "Towards a weakly supervised framework for 3d point cloud object detection and annotation" for 3D AVS object detection and "Exploring cross-image pixel contrast for semantic segmentation", "Rethinking Semantic Segmentation: A Prototype View" for scene analysis.

It is not clear the meaning of the "outcomes" column in the tables. The outcomes of different methods are not consistently measured with a unified metric. This makes it hard to compare the methods from the "outcomes".

Author Response

Response to Reviewer – 03

  1. A detailed taxonomy of the algorithms in terms of tree or figure should be provided. That will help understanding how different algorithms or tasks can be tied together.

Response 1. Thank you for your kind suggestions to improve this paper. Taxonomy of the algorithms from each domain and subdomains are shown in Figure 4 as suggested to provide a clear summarized view to the reader.

  1. Some recent advances in the field should be also discussed. For example, "Towards a weakly supervised framework for 3d point cloud object detection and annotation" for 3D AVS object detection and "Exploring cross-image pixel contrast for semantic segmentation", "Rethinking Semantic Segmentation: A Prototype View" for scene analysis.

Response 2. I appreciate your kind suggestion. The resources you have shared were beneficial and knowledgeable. The "Exploring cross-image pixel contrast for semantic segmentation" article was reviewed in this review paper. However, the "Towards a weakly supervised framework for 3D point cloud object detection and annotation" article applied LiDAR as the primary sensor. However, the theme of this review paper is RGB vision-based approaches. Furthermore, the "Rethinking Semantic Segmentation: A Prototype View" article was published in 2022, but the year ranged in this paper is from 2011 to 2021. However, some recent advanced articles in the field were newly added (for instance, [21], [34], [110], [111], [147], and [150] were added to review and discussed in Table 2).

  1. It is not clear the meaning of the "outcomes" column in the tables. The outcomes of different methods are not consistently measured with a unified metric. This makes it hard to compare the methods from the "outcomes".

Response 3. The column "outcomes" means the results, performance or even significance of the reviewed paper from the domain or subdomain. Although it is tried to our level best maintain same evaluation metrics, as the core flow and focus discussing with advantages and disadvantages and covering some important criteria or discuss a solution based on previous discussed papers' limitations. Despite having different methods, most of them followed a pattern. In the discussion or end of the paragraph, a discussion is included to clarify the reader regarding the performance and to analyse how the method works or the most suitable methods.

Round 2

Reviewer 1 Report

Aince the authors have incorporated moat of the concerns raised. The manuscript seems fine for acceptance